

# The Caltech Photooxidation Flow Tube Reactor - I: Design and Fluid Dynamics

Y. Huang[1,***], M. M. Coggon[2,*,***], R. Zhao[2], H. Lignell[2,**], M. U.  Bauer[2], R. C. Flagan[1,2], and J. H. Seinfeld[1,2]

[1]Department of Environmental Science and Engineering, California Institute of Technology, Pasadena, CA, USA
[2]Division of Chemistry and Chemical Engineering, California Institute of Technology, Pasadena, CA, USA
[*]Now at CIRES, University of Colorado, and NOAA Earth System Research Laboratory, Boulder, CO, USA
[**]Now at South Coast Air Quality Management District, Diamond Bar, CA, USA
[***]These authors contributed equally to this work.

*Correspondence to:* J. H. Seinfeld (seinfeld@caltech.edu)

**Abstract.** Flow tube reactors are employed to study gas-phase atmospheric chemistry and secondary organic aerosol formation. A new laminar flow tube reactor, the Caltech PhotoOxidation flow Tube (CPOT), has been designed with the aim of achieving a well-characterized fluid dynamic and residence time environment. We present here the design and fluid dynamical characterization of the CPOT, based on the fundamental behavior of vapor molecules and particles in the reactor. The design

of the inlet of the CPOT, which was based on computational fluid dynamics (CFD) simulations, comprises a static mixer and a conical diffuser to facilitate rapid development of the characteristic laminar flow parabolic profile. A CFD laminar flow model is developed to simulate the residence time distribution (RTD) of vapor molecules and particles in the CPOT. To assess the extent to which the actual performance adheres to the theoretical CFD model, RTD experiments were conducted with $O_3$ and sub-micrometer ammonium sulfate particles. The measured RTD profiles do not strictly adhere to theory, owing to slightly

non-isothermal conditions in the reactor, which lead to secondary flows. Introducing an enhanced eddy-like diffusivity for the vapor molecules and particles in the laminar flow model significantly improves the model-experiment agreement. These characterization experiments, in addition to the idealized computational behavior, provide a basis on which to evaluate the performance of the CPOT as a chemical reactor.

## 1   Introduction

Flow tube reactors are an alternative to the batch chamber as a means to study atmospheric chemistry and secondary organic aerosol formation. The flow tube reactor is generally operated under steady-state conditions. An attribute of the flow tube reactor is that, by control of the inlet concentrations and photolytic conditions, it is possible to generate hydroxyl (OH) radical concentrations that exceed substantially both those in the atmosphere and those generally achievable in batch chambers (Kang et al., 2007, 2011; Kroll et al., 2009; Smith et al., 2009; Lambe et al., 2011a, 2012, 2015; Keller and Burtscher, 2012; Slowik





et al., 2012; Chen et al., 2013; Li et al., 2015; Peng et al., 2015, 2016), affording the study of atmospheric oxidation under conditions equivalent to multiple days of atmospheric OH exposure. For example, the Potential Aerosol Mass (PAM) reactor, among the first of the modern generation of atmospheric flow tube reactors, can achieve OH exposures (concentration × time) from $2 \times 10^{10}$ to $1.8 \times 10^{12}$ molec cm$^{-3}$ s (Kang et al., 2007, 2011), at which it is possible to simulate atmospheric reaction conditions comparable to those occurring over $\sim$ 1week. Flow reactor systems have been used for a variety of secondary organic aerosol formation and aging studies (Kroll et al., 2009; Smith et al., 2009; Ezell et al., 2010; Lambe et al., 2011a, b, 2012, 2015; Ortega et al., 2013; Chen et al., 2013; Bruns et al., 2015; George et al., 2007; Keller and Burtscher, 2012). Modeling studies have investigated the radical chemistry in the oxidation flow reactor (Li et al., 2015; Peng et al., 2015, 2016).

Flow tube designs vary in dimensions, detailed construction, and strategy for generating the oxidizing environment. Each specific design aspect of a reactor can significantly affect both the flow dynamics and the chemistry within the reactor. An approach combining theoretical, modeling, and experimental methods is useful to fully understand the behavior of a flow tube reactor at the fundamental level.

We describe here the design and characterization of the Caltech PhotoOxidation flow Tube (CPOT). We highlight fundamental considerations of the fluid dynamics within a laminar flow tube reactor, including methods of injection, behavior of vapor and particles, as well as the effects of non-isothermal conditions on the fluid dynamics. We incorporate and examine these considerations through a modeling framework that employs computational fluid dynamics (CFD) simulations. Finally, we evaluate the model framework by comparison with experimental measurements.

## 2 Design and Experimental Setup

### 2.1 CPOT Reactor

The CPOT comprises three sections: the inlet section, the main reaction section, and the outlet section (Fig. 1A). The inlet consists of two components - the static mixer and the conical diffuser (Fig. 1B). The static mixer is designed to thoroughly mix reactant streams, whereas the diffuser serves to expand the mixed flow to the diameter of the reaction section while maintaining an idealized laminar flow profile. The static mixer is constructed of stainless steel and consists of 12 helical elements (StaticMixCo, NY). The Pyrex glass diffuser section expands from an inner diameter of 1.6 cm to 15 cm at an angle of 15°. The diffuser angle was chosen based on CFD simulations in order to minimize flow separation and recirculation. Detailed design of the inlet section is discussed in Section 3.

The CPOT reaction section consists of two 1.2 m × 17 cm ID cylindrical quartz tubes surrounded by an external water jacket (1 cm thickness) and flanged together with clamps and chemically resistant o-rings. Four ports along the reactor axis allow sampling of the reactor contents at different residence times. A transition cone at the end of the reactor concentrates the reactants into a common sampling line that can be split among multiple instruments; thus, samples extracted at the end of the reactor represent the so-called cup-mixed average of the entire reactor cross section. This design is similar to the exit cone of the UC Irvine flow tube reactor (Ezell et al., 2010). The Pyrex glass exit cone gradually reduces the diameter of the reactor from 15 cm to 0.72 cm at an angle of 15°. Similar to the inlet diffuser, the exit cone is temperature-controlled.





The CPOT is designed to operate under laminar flow. The essential dimensionless group that differentiates laminar vs. turbulent flow is the Reynolds number, $\mathrm{Re} = \dfrac{\rho U D}{\mu}$, where $\rho$ is the fluid density, $U$ is a characteristic velocity of the fluid, $\mu$ is the fluid viscosity, and $D$ is the tube diameter. For cylindrical tubes, the flow is considered laminar when $\mathrm{Re} < 2100$. Under the typical CPOT flow rate (2 L min$^{-1}$), the maximum Reynolds number ($\mathrm{Re} = 450$) in the reactor occurs at the exit cone, well

below the transition to turbulent flow.

## 2.2 Photolytic Environment

The reactor is housed within a $51 \times 51 \times 300$ cm chamber containing 16 wall-mounted UV lamps. The arrangement of the lamps is outlined in Fig. 1D. Light intensity is adjustable, and the UV spectrum can be set to a specific wavelength range with the installation of various T12 UV lamps, including Hg vapor lamps (irradiate mainly at 254 nm), UVB lamps (polychromatic

irradiation centered at 305 nm) and UVA lamps (polychromatic irradiation centered at 350 nm). The water coolant in the jacket surrounding the tube is transparent at the UV wavelengths of interest, with the exception that it absorbs at the 185 nm band emitted by the Hg vapor lamps. Although the general UV cutoff of water is at 190 nm, we observed formation of $\sim$50 ppb of $O_3$ in the CPOT with a 2 L min$^{-1}$ flow rate under the full power of the Hg vapor lamp, indicating that some 185 nm radiation penetrates into the reaction section. Quantification of light fluxes for each type of lamp is the prerequisite for performing

photochemical experiments in the CPOT.

## 2.3 Temperature Control in the Reaction Section

At full photolytic intensity, the lamps generates a substantial amount of energy. To maintain a constant temperature and minimize convective mixing in the tube due to temperature inhomogenity, each of the two reaction sections is fitted with a quartz cooling jacket, in which chilled water is circulated at a rate of 13 L min$^{-1}$. Coolant is introduced into the jacket near the exit

cone and exits at the inlet (Fig. 1A). Under typical operation, the cooling jacket can maintain the steady-state reactor temperature at a desired value between 20 and 38°C. Under full photolytic intensity of the UVA lamps, which produce the most heating among the three types of lamps, the temperature rise of air in the reactor is $\leq 0.3$ K at steady state. Reactor temperature control is addressed in Section 3.3.

## 2.4 Experimental Testing

Particles and vapor species introduced to the CPOT were used to characterize the fluid dynamics experimentally. The injection scheme is illustrated by Fig. 1C. Polydisperse ammonium sulfate particles were generated by atomizing a 0.01 M aqueous solution with a constant rate atomizer (Liu and Lee, 1975). The atomized particles were immediately dried by a silica gel diffusion drier. The size distribution of particles was measured by a custom-built scanning mobility particle sizer (SMPS). For the particle residence time distribution (RTD) measurement, the particle counts were monitored with a TSI 3010 Condensation

Particle Counter (CPC). Gas-phase RTD studies were performed under dark condition. $O_3$ was generated by passing purified air through an $O_3$ generator (UVP, 97-0067-01), and the $O_3$ mixing ratio was monitored by a $O_3$ monitor (Horiba APOA-360).



## 2.5 Computational Fluid Dynamics (CFD) Simulations

CFD simulations were performed using COMSOL Multiphysics 5.0 software (Stockholm, Sweden. www.comsol.com) to assist the design and characterization of the CPOT. COMSOL uses a finite element method to solve transport problems and has many built-in modules that can be utilized to simulate a specific experimental condition. Recently, several research groups have

5 employed COMSOL in atmospheric and aerosol chemistry studies (Grayson et al., 2015; Sellier et al., 2015; Zhang et al., 2015). Here, the model geometry replicates that of the actual design; thus, the simulations include a static mixer, diffuser inlet, reaction section, and exit cone with dimensions identical to those of the CPOT (Fig. 2A).

At the design stage, the performance of the inlet section was simulated numerically using CFD models (Section 2.5). The actual static mixer containing 12 mixing element was simulated by a two-element mixer using the COMSOL built-in static

mixer model. Flow profiles calculated using the 2-element static mixer model were found to be identical for simulations conducted using static mixers with 4 or more elements. Since static mixers yield asymmetric flow patterns, the model was solved in a 3D geometry. The entire 3D model was discretized with a fine mesh composed of approximately $1.25 \times 10^6$ tetrahedral elements (Fig. 2B). The average element quality, which is a reflection of cell distortion (a value of 1 reflects a perfect element shape), was 0.77 with a minimum of 0.12. A finer mesh within the domain of the static mixer was applied to capture flow

dynamics near the entrance to the diffuser cone (Fig. 2B). Model sensitivity to meshing was tested using a finer mesh density, and results were found to be identical. An impermeable and no-slip boundary condition was applied to all surfaces. The flow at the entrance into the static mixer was set to be 2 L min$^{-1}$, and the outlet pressure was assumed to be atmospheric. Simulations were conducted until a steady state was achieved, and the errors converged to $< 10^{-6}$.

Navier-Stokes equations were solved using the COMSOL laminar flow package in the CFD module assuming compressible,

isothermal flow. To evaluate the effect of reactor temperature gradients, the COMSOL laminar flow package was coupled to the convective and diffusive heat transfer interface. To visualize fluid flow through the reactor, transient simulations were performed using the COMSOL dilute species transport package. This model, when coupled to the Navier-Stokes equations, enables one to track convection and diffusion of a tracer species, as described in Section 4. After first generating the steady-state laminar flow profile, a 30 s rectangular pulse of a 0.1 mol m$^{-3}$ tracer was introduced numerically into the reactor at the entrance to the static

mixer to generate the RTD. The wall was assumed impenetrable for the RTD simulation. Molecular or Brownian diffusivity can be varied over several orders of magnitude to represent that of vapor molecules and particles. The simulation was run for 80 min with data output every 15 s (consistent with the data acquisition of the instrument). Simulations were performed for a variety of different inlet geometries, flow rates, and reactor temperature gradients. These simulations served to evaluate the CPOT design against alternative configurations and also demonstrate the sensitivity to various flow conditions.

## 30  3  Design of the Flow Tube Reactor

Essential elements of the design of a flow tube reactor are: (1) the manner by which reactants are introduced into the reactor; (2) the nature of the flow inside the reactor; (3) the type and location of the radiation source relative to the reactor itself; and (4) the management of heat generation owing to the radiation source. The first two correspond to the inlet section design, while





the latter two address the problem of possible non-isothermal conditions in the reaction section. In this section, the design of the CPOT is discussed.

## 3.1 Injection Method

A number of possible arrangements exist to introduce material into a flow tube reactor (Fig. 3). The nature of the injection

manifold has the potential to profoundly affect the flow profile in the subsequent reaction section. In the case of a laminar flow reactor, it is desirable to minimize such "end effects" in order to establish parabolic flow quickly within the reaction section; otherwise, phenomena such as jetting and recirculation have the potential to impact flow patterns throughout the entire reactor. Figure 3A depicts the injection method utilized by the PAM (Kang et al., 2007; Lambe et al., 2011a) reactor, by which vapor and particles are introduced into the reaction section through a short injection tube. While a benefit of this design is its

simplicity, with this mode of injection, it is challenging to distribute reactant mixtures evenly across the reactor cross section. We tested this inlet method on a cylindrical Pyrex glass tube and visualized the flow pattern by the injection of smoke (Fig. 3A). With flow controlled by a vacuum line attached to the exit section, the gas-particle mixture is pulled into the reaction tube at a rate that is dictated by mass conservation. Smoke visualization studies illustrate that the mixture concentrates in a plug at the center of the reactor. This "fire hose" effect arises from the enhanced velocity at the exit of the injection tube ($U_{\text{avg, injection}}$).

At an overall flow rate of 1 L min$^{-1}$, the average velocity exiting a standard 6.35 mm entrance tube (ID = 3 mm) is 2.35 m s$^{-1}$. This is ~2500 times the average velocity within the reactor ($U_{\text{avg, bulk}}$=0.09 cm s$^{-1}$). As discussed by Lambe et al. (2011a), this injection method has the potential to induce dead volume near the entrance of the reaction section and promote reactor-scale recirculation. Such flow behavior is typical for that occurring with a sudden expansion (Bird et al., 2007). In the new version of PAM, a larger inlet is employed to reduce recirculation in the reactor (Ortega et al., 2016).

Some flow tube designs address these inlet issues using flow management devices. The UC Irvine flow tube reactor (Ezell et al., 2010) utilizes a spoked-hub/showerhead disk inlet that distributes the reactants evenly about the reactor cross-section and provides sufficient mixing (Fig. 3B). With a showerhead disk, the reactants can be mixed and introduced into the tube in a controlled manner. Even when reactants are introduced gently into the tube, an axial distance is still required for the flow to develop to the characteristic parabolic laminar flow profile. This entrance length, $L_{\text{entr}}$, is estimated to be $0.035D\text{Re}$ (Bird et al.,

2007). Ezell et al. (2010) designed the inlet with a sufficient entrance length $L_{\text{entr}}$ to ensure the development of the laminar profile prior to the reaction section.

In the CPOT, reactants are injected via a conical diffuser (Fig. 3C) which has the advantage of gradually decreasing in the velocity, thereby assisting with the formation of the laminar parabolic profile. The employment of a diffuser cone essentially replaces $L_{\text{entr}}$, and a parabolic profile is fully developed when the reactants reach the reaction section.

## 30  3.2 Angle of the Diffuser

The CPOT employs a conical diffuser and a static mixer. A key consideration in designing a diffuser is avoiding flow separation that occurs when streamlines detach from the diffuser wall. Separation may be characterized by two flow patterns: stall and jetting flow. In stall, an asymmetrical flow pattern develops due to an adverse pressure gradient. Fluid is accelerated along one



wall of the diffuser and recirculates slowly back along the other wall to the point of streamline detachment (Tavoularis, 2005). As demonstrated in Fig 3A, jetting flow is characterized by a symmetric flow pattern where fluid is accelerated at the center of the diffuser and recirculates slowly along the walls. Recirulation introduces non-ideality since it accelerates gases and particles down the reactor, thereby affecting the RTD and leading to uncertain reaction times.

Diffusers are routinely applied in larger systems such as wind tunnels and turbines; therefore, most literature on diffuser design focuses on flow patterns at high Re (Re > 5000, e.g. Mehta and Bradshaw, 1979; Seltsam, 1995; Tavoularis, 2005; Prakash et al., 2014). As a rule of thumb for high Re systems, flow separation can be suppressed if the diffuser half-angle is $\leq 5°$; however, smaller angles are needed if the area ratio between the diffuser inlet and reactor section is much greater than 5 (Mehta and Bradshaw, 1979). We are unaware of studies that report diffuser performance at modest Re (< 500). Fried and
Idelchik (1989) recommend that diffusers be designed with an angle of divergence < 7° to avoid flow separation; alternatively, White (2008) recommends an angle < 15°. Sparrow et al. (2009) modeled the flow of fluid through diffuser cones at various Re. For further discussion about flow separation within diffusers, see Tavoularis (2005).

Under a typical CPOT working flow rate (2 L min$^{-1}$), the value of Re at the entrance of the conical diffuser is ∼ 200. Figure 4 shows simulated flow profiles for a range of diffuser angles. The red traces represent streamlines, whereas the blue surface
illustrates points where flow recirculation occurs, that is, where the axial velocity $< 0\,\mathrm{cm\,s^{-1}}$. Collectively, these traces provide a visualization of the recirculation zone. We present flow profiles in the presence and absence of a static mixer since swirling flow has been shown to improve diffuser performance for systems with appreciable separation (McDonald et al., 1971).

As the diffuser angle increases, separation becomes more appreciable, and the recirculation zone penetrates farther into the reaction section ($\Delta z > 0$). At the most extreme angle we considered ($\theta = 37°$), the simulation predicts that the first 46
20   cm of the reaction section is impacted by recirculation. For reference, the extreme of a sudden expansion ($\theta = 90°$) exhibits recirculation that penetrates nearly halfway through the reactor ($\Delta z = 110$ cm). For flow tube systems operated at similar Re as here, if a parabolic flow profile is desired, it is recommended that one utilize a diffuser with $\theta < 20°$ in order to minimize laminar flow disturbance within the reaction section.

The presence of a static mixer tends to quell separation at moderate diffuser angles. The recirculation zone appears to prop-
agate into the reaction section only at diffuser angles $> 30°$; however, the extent of this recirculation is substantially reduced compared to simulations in the absence of a static mixer. Furthermore, the recirculation zone is predicted to be symmetric; fluid from the static mixer is directed radially towards the walls of the diffuser and recirculates back towards the center. In contrast, the recirculation zone in diffusers without static mixers is predicted to be asymmetric (see Fig. 4), with flow recirculating at one wall of the diffuser. With flow introduced via a sudden expansion, the presence of a static mixer does little to minimize
recirculation. The improvement in diffuser performance with swirling flow at the inlet is consistent with observations at high Re (McDonald et al., 1971), suggesting that the addition of a static mixer may help to mitigate moderate separation in systems employing wide-angled diffusers.

Figure 5 further illustrates the CFD-modeled velocity profiles for the actual CPOT design, with a 15° diffuser cone coupled to a static mixer, in the region of the reactor extending from the inlet cone to the first 10 cm of the reaction section. We refer to this
section of the reactor as the "inlet-affected" region, since axial positions farther downstream exhibit fully-developed laminar



profiles. Figure 5A visualizes the entire velocity field along select cross-sections within the inlet-affected region, whereas Fig. 5B presents 1D velocity profiles at various axial positions. Note that Fig. 5A presents the velocity magnitude, whereas Fig. 5B illustrates the axial velocity component (i.e., flow in the $z$-direction) to facilitate identification of regions impacted by flow recirculation. In general, the simulation predicts the absence of recirculation within the reactor under isothermal conditions.

As demonstrated by Fig. 5B, the simulated axial velocity profile immediately downstream of the static mixer exhibits two jets with a maximum axial velocity of 31 cm s$^{-1}$. The jets quickly dissipate as the flow develops through the diffuser cone. At the exit of the diffuser cone, the flow is nearly parabolic, and the maximum velocity slows to 0.5 cm s$^{-1}$. Within 10 cm of the diffuser exit, the flow becomes parabolic with a maximum centerline velocity of 0.4 cm s$^{-1}$. We also simulated the fluid field under higher flow rates (e.g. 4 and 6 L min$^{-1}$) and found no separation of flow within the diffuser. These results demonstrate

that the CPOT inlet is within the design limits for a diffuser with non-separated flow and that the presence of a static mixer has little effect on the parabolic profile in an isothermal reaction section. However, the presence of the static mixer does have an impact on the residence time of the reactants. Fig. 5C shows the corresponding residence time at the three positions for both vapor molecules and monodisperse particles, both of which are 30 s square wave input. Section 4.3 addresses RTD.

### 3.3 Non-isothermal Effect

Precise control of temperature is crucial in maintaining as well-characterized a laminar flow as possible in the reaction section (Khalizov et al., 2006). In one class of flow tube design, the radiation source is positioned within the flow tube reactor itself, and the reactor walls are constructed of a UV blocking material. In that design, the effect of the internal heat source on the flow must be considered. In the present design, with the reaction tube suspended at the center of the chamber and the lights positioned on the outside of the tube, an exterior water jacket provides a heat transfer medium, while allowing penetration of

UV radiation to the reactor. If water recirculation in the jacket is sufficiently rapid, axial temperature gradients in the cooling jacket can be minimized. Any jacket temperature maintained appreciably below or above that in the reactor itself will lead to temperature gradients that may induce the secondary flows in the reactor.

Although the CPOT is equipped with a temperature control system (Section 2.3), maintaining a target temperature under UV irradiation is challenging. The measured rise in coolant temperature in the CPOT at steady state under full irradiation

conditions is within $\leq 0.2$ K. Given the absence of heat sources within the reactor itself, the increase in coolant temperature is a result of the absorption of heat generated by the exterior UV lamps. Temperature gradients along the reactor wall have the potential to induce recirculation if changes in density cause free convection flow within the tube. The establishment of radial temperature gradients near the wall induces recirculation cells as density variations force the flow to stratify.

The dimensionless group that characterizes the effect of free convection on flow is the Richardson number, which relates the

30 strength of buoyancy forces to that of convective forces. The Richardson number (Holman, 2010), Ri, can be expressed as the ratio of the Grashof number, Gr, to the square of the Reynolds number, Re:

$$\text{Ri} = \frac{\text{Gr}}{\text{Re}^2} = \frac{g\beta D^3 \Delta T / \nu^2}{(\rho U_{\text{avg}} D / \mu)^2} \sim \frac{gD}{TU_{\text{avg}}^2} \Delta T \tag{1}$$





where $g$ is the gravitational acceleration, $\beta$ is the thermal expansion coefficient of air ($\frac{1}{T}$ for ideal gases), $U_{\mathrm{avg}}$ is the average fluid velocity, $\nu$ is the kinematic viscosity of air ($\frac{\mu}{\rho}$), and $\Delta T$ is a characteristic temperature difference between the tube wall and centerline. When $\mathrm{Ri} < 0.1$, convective forces dominate, and effects of buoyancy on the flow are small. When $\mathrm{Ri} > 10$, buoyancy forces may lead to flow bifurcation and recirculation. Under typical CPOT operating conditions, a radial temperature

gradient between the fluid and wall of $\sim 0.007$ K is required to maintain $\mathrm{Ri} < 10$. Because this is a very small temperature difference, modest inequalities in temperature are anticipated to affect flow patterns within the reactor.

     To investigate the effect of wall temperature differences on flow within the reaction section, we performed COMSOL simulations, assuming a non-isothermal reactor wall. The COMSOL laminar flow package was coupled to the convective and diffusive heat transfer interface. Since water is assumed to flow uniformly through the annular water jacket cross section from

the exit to the entrance, we apply an axial temperature gradient to the simulation. A schematic illustration of the simulation setup is shown in Fig. 2C. At the exit of the reaction section, the temperature of the reactor wall is set to that of water entering the cooling jacket ($T_{\mathrm{in}}$). At the entrance to the reaction section, we assume that the wall temperature is that of the water exiting the cooling jacket ($T_{\mathrm{out}} = T_{\mathrm{in}} + \Delta T$). The wall temperature is assumed to change linearly between the entrance and exit. The diffuser cone is prescribed at a constant temperature equivalent to the cooling jacket temperature $T_{\mathrm{out}}$, whereas the exit cone is

prescribed a constant temperature of $T_{\mathrm{in}}$. In the following discussion, we focus on results with $T_{\mathrm{in}} = 23\,^{\circ}\mathrm{C}$, which is the typical room temperature in the Caltech laboratory. Note that this model setup is a simplified case, since in actual experiments the entrance and exit cones should be kept at the same temperature (i.e., $T_{\mathrm{in}}$); this will introduce temperature discontinuity between the entrance cone and the reaction tube. Nonetheless, this idealized model provides insight into the temperature difference induced flow perturbation within the flow tube reactor.

Figure 6 demonstrates the simulated effect of an axial temperature gradient ($\Delta T$) on flow profiles within the CPOT. Figure 6A illustrates 2D velocity profiles at various axial positions and a blue isosurface where the axial velocity $< 0 \ \mathrm{cm \ s^{-1}}$. Figure 6B illustrates 1D velocity profiles at the midpoint of the reactor. As the temperature gradient within the reactor increases, the velocity profiles skew owing to buoyancy of the warm air. This bifurcation induces recirculation and is predicted to affect the entire reactor region. For a temperature gradient of 0.2 K (equivalent to that measured for the CPOT), the recirculation zone

exhibits a maximum velocity of $-0.15 \ \mathrm{cm \ s^{-1}}$.

     The simulations demonstrate the sensitivity of the velocity profile in the reactor to small temperature gradients within the reaction section. Such disturbances will manifest in shorter, broader residence times due to induced recirculating flow. As demonstrated in Fig. 6, a critical temperature difference exists at which recirculation becomes important. At a volumetric flow of $2 \ \mathrm{L \ min^{-1}}$, this critical temperature difference between the exit and the entrance is estimated to be $\sim 0.08$ K. The Ri

number criterion indicates that higher flow rates reduce the reactor sensitivity to temperature gradients. We find the critical temperature differences at $4 \ \mathrm{L \ min^{-1}}$ and $6 \ \mathrm{L \ min^{-1}}$ are 0.18 K and 0.3 K, respectively. Even at higher flow rates, relatively small temperature gradients along the walls of the reactor can have potential consequences on reactor flow patterns.





## 4 Behavior of Gaseous and Particulate Species in a Laminar Flow Tube Reactor

### 4.1 Fluid Field in the Reaction Section

The CPOT is designed ideally to be operated under isothermal laminar flow conditions. At the typical operating flow rate (2 L min$^{-1}$), flow within the reaction section is laminar (Re $\cong$ 20), under which the axial fluid velocity is given by the parabolic distribution,

$$v_z(r) = U_{\max}\left[1 - \left(\frac{r}{R}\right)^2\right] \tag{2}$$

where $U_{\max}$ is the centerline velocity, $r$ is the radial coordinate in the tube, and $R$ is the tube radius. The residence time of fluid elements in laminar flow differs along streamlines, for which the average residence time of fluid elements is precisely calculated. Owing to the sensitivity to small temperature difference, as noted above, the actual velocity profile in the reactor may not adhere to an ideal parabolic distribution. (In contrast, the average velocity profile in turbulent flow is essentially uniform across the tube; however, the transport of material to the wall is significantly enhanced.)

### 4.2 Penetration Efficiency ($\eta$)

The full conservation equation for gas-phase species $i$, $c_i(t, r, z)$, is

$$\frac{\partial c_i}{\partial t} + v_z(r)\frac{\partial c_i}{\partial z} = \mathscr{D}_i\left[\frac{1}{r}\frac{\partial}{\partial r}\left(r\frac{\partial c_i}{\partial r}\right) + \frac{\partial^2 c_i}{\partial z^2}\right] + P_i - S_i \tag{3}$$

where $\mathscr{D}_i$ is the molecular diffusivity of species $i$ in air, and $P_i$ and $S_i$ are the rates of generation and consumption of species $i$, by chemical reaction and gas-particle partitioning, respectively. A typical order of magnitude of the molecular diffusion coefficient for vapor molecules in air is $\sim 10^{-5}$ m$^2$ s$^{-1}$. Under conditions typical of the operation of the flow tube reactor, the magnitude of the axial molecular diffusion term in Eq. (3) is small relative to that of the other terms in the equation and may be neglected. When the system is at steady-state without generation and consumption of species $i$, Eq. (3) becomes,

$$v_z(r)\frac{\partial c_i}{\partial z} = \mathscr{D}_i\left[\frac{1}{r}\frac{\partial}{\partial r}\left(r\frac{\partial c_i}{\partial r}\right)\right] \tag{4}$$

Equation 4 is subject to a boundary condition at the reactor entrance, $z = 0$:

$$c_i(r, 0) = c_{i0} \tag{5}$$

where a uniform concentration $c_{i0}$ is assumed at the inlet of the reactor, and the symmetry condition at the centerline of the reactor, $r = 0$:

$$\frac{\partial c_i}{\partial r}(0, z) = 0 \tag{6}$$

A general boundary condition at the reactor wall allows for the possible deposition of species $i$ on the wall, is

$$\mathscr{D}_i\frac{\partial c_i}{\partial r}(R, z) = -k_{wi}c_i \tag{7}$$



where $k_{wi}$ is a first-order wall deposition coefficient for species $i$. $k_{wi}$ can be expressed in terms of the uptake coefficient for species $i$, $\gamma_i$, as $k_{wi} = \frac{1}{4}\gamma_i \bar{c}_i$, where $\bar{c}_i$ is the mean molecular speed of species $i$. Either $k_{wi}$ or $\gamma_i$ must be determined experimentally. If no uptake of species $i$ occurs at the wall, then $k_{wi} = 0$.

The penetration efficiency $\eta$ is defined as the fraction of material entering the reactor that leaves in the absence of chemical reaction. If no removal occurs during flow through the reactor, then $\eta = 1$. Equations (4) to (7) can be solved either numerically or analytically to determine the penetration efficiency $\eta$, given $k_{wi}$. Davis (2008) presented an analytical solution of Eqs. (4) to (7). For the case of complete removal of species $i$ at the wall, in which $k_w \to \infty$, corresponding to $c_i = 0$ at the wall, the analytical solution for $\eta$ is (Fuchs, 1964):

$$\eta = 0.8191\exp(-3.657\xi) + 0.0975\exp(-22.3\xi) + 0.0325\exp(-57\xi) + \dots \tag{8}$$

where $\xi$ is the dimensionless length ($\frac{\pi\mathscr{D}L_{\mathrm{cyld}}}{Q}$), $L_{\mathrm{cyld}}$ is the length of the cylindrical tube, and $Q$ is the volumetric flow rate. For small $\xi$, i.e. $< 0.02$, an alternative equation is recommended (Gormley and Kennedy, 1948):

$$\eta = 1 - 2.56\xi^{2/3} + 1.2\xi + 0.177\xi^{4/3} \tag{9}$$

The penetration efficiency for particles is size dependent, i.e. $\eta(D_p)$, since in addition to convection by the fluid field and Brownian diffusion, particles undergo gravitational settling. We will address this together with the RTD of particles in Section 5.1.2.

## 4.3 Residence Time Distribution (RTD)

In a laminar flow field, idealized non-diffusing vapor or non-diffusing and non-settling particles, introduced as a pulse at the entrance of the tube, will first emerge as a pulse at the residence time of the centerline, followed by a decaying curve as the material on the slower streamlines reaches the exit. Under actual conditions, vapor molecules undergo molecular diffusion in both the radial and axial directions, and particles are subject to Brownian diffusion and gravitational settling.

### 4.3.1 Vapor Molecule Residence Time Distribution

Vapor molecules in laminar flow in a tube undergo molecular diffusion in both the radial and axial directions. With molecular diffusion coefficient $\mathscr{D}_i$, the characteristic diffusion time in the radial direction is $\tau_{c,\mathscr{D}_i} = \frac{R^2}{\mathscr{D}_i}$. To assess the importance of radial diffusion as a mechanism for smearing vapor molecules across the tube cross-section during convection down the tube, one can compare the characteristic timescale for radial diffusion with the characteristic residence time in the cylindrical tube, $\tau_{c,\mathrm{cyld}} = \frac{L_{\mathrm{cyld}}}{U_{\mathrm{avg}}}$. If $\tau_{c,\mathscr{D}_i} \ll \tau_{c,\mathrm{cyld}}$, for example, the vapor molecules will diffuse more or less uniformly across the tube radius in the time it takes for the fluid to flow to the tube exit. Likewise, if $\tau_{c,\mathscr{D}_i} \gg \tau_{c,\mathrm{cyld}}$, each vapor molecule will effectively remain on the streamline upon which it entered. Vapor molecules also diffuse in the axial direction; this process is represented by the axial diffusion term, $\mathscr{D}_i \frac{\partial^2 c_i}{\partial z^2}$, on the right hand side of Eq. (3). As noted earlier, for flow velocities of the magnitude of those in the CPOT, the effect of this axial diffusion is negligible when compared with axial convection. However, an apparent axial diffusion can arise from the interaction of radial molecular diffusion and the laminar flow, a process known as Taylor dispersion




(Taylor, 1953; Bird et al., 2007). Under the criterion, $\tau_{c,\text{cyld}} \gg \frac{\tau_{c,\mathcal{D}_i}}{3.83^2}$, the concentration becomes approximately uniform over the cross-section of the tube. By cross-section averaging of Eq. (3) (without sources and sinks), the average concentration at any cross section obeys:

$$\frac{\partial \langle c_i \rangle}{\partial t} + U_{\text{avg}} \frac{\partial \langle c_i \rangle}{\partial z} = \mathcal{D}_{\text{eff},i} \frac{\partial^2 \langle c_i \rangle}{\partial z^2} \tag{10}$$

where $\mathcal{D}_{\text{eff},i} = \mathcal{D}_i \left( 1 + \frac{\text{Pe}^2}{192} \right)$ which accounts for the convective enhancement in diffusivity (Aris, 1956), where Pe, the Péclet number, is $\frac{2RU_{\text{avg}}}{\mathcal{D}_i}$.

The solution of Eq. (10) for a pulse input at the entrance to the reactor, of $N_0$ moles over the cross section area $A$ of the tube, is:

$$\langle c_i \rangle(t,z) = \frac{1}{\sqrt{4\pi \mathcal{D}_{\text{eff},i} t}} \frac{N_0}{A} \exp\left[ -\frac{(z - U_{\text{avg}}t)^2}{4\mathcal{D}_{\text{eff},i} t} \right] \tag{11}$$

The RTD of the diffusive species in the flow tube, i.e. at $z = L_{\text{cyld}}$, is:

$$\langle c_i \rangle(t, L_{\text{cyld}}) = \frac{1}{\sqrt{4\pi \tilde{\mathcal{D}}_{\text{eff},i} \tilde{t}}} \frac{N_0}{V} \exp\left[ -\frac{(1 - \tilde{t})^2}{4\tilde{\mathcal{D}}_{\text{eff},i} \tilde{t}} \right] \tag{12}$$

where $V$ is the volume of the tube, $\tilde{\mathcal{D}}_{\text{eff},i} = \frac{\mathcal{D}_{\text{eff},i}}{\tau_{c,\text{cyld}} U_{\text{avg}}^2}$, and $\tilde{t} = \frac{t}{\tau_{c,\text{cyld}}}$.

For a pulse input, of finite duration $t_0$,

$$\langle c_i \rangle(t, z=0) = \begin{cases} \dfrac{N_0}{A U_{\text{avg}} t_0} & 0 \le t \le t_0 \\ 0 & t > t_0 \end{cases} \tag{13}$$

The RTD at $z = L_{\text{cyld}}$ is:

$$\langle c_i \rangle(t, L_{\text{cyld}}) = \frac{N_0}{2V} \left[ \text{erf}\left( \frac{1 - \tilde{t}}{\sqrt{4\tilde{\mathcal{D}}_{\text{eff},i} \tilde{t}}} \right) - \text{erf}\left( \frac{1 - \tilde{t} - \tilde{t}_0}{\sqrt{4\tilde{\mathcal{D}}_{\text{eff},i} \tilde{t}}} \right) \right] \tag{14}$$

where $\text{erf}(x) = \frac{2}{\sqrt{\pi}} \int\limits_0^x \exp(-\eta^2) d\eta$ and $\tilde{t}_0 = \frac{t_0}{\tau_{c,\text{cyld}}}$. More generally, by transforming $t = -\frac{z}{U_{\text{avg}}} = -\frac{z}{L_{\text{cyld}}} \frac{L_{\text{cyld}}}{U_{\text{avg}}} = -\tilde{z}\tau_{c,\text{cyld}}$,

where $\tilde{z} = \frac{z}{L_{\text{cyld}}}$, the RTD for a non-ideal pulse input $f(t)$ (e.g. the solid profile in Fig. 5C) is in the form of:

$$\langle c_i \rangle(t, L_{\text{cyld}}) = \frac{1}{\sqrt{4\pi \tilde{\mathcal{D}}_{\text{eff},i} \tilde{t}}} \frac{N_0}{V} \int\limits_{-\infty}^{+\infty} f(-\tilde{z}\tau_{c,\text{cyld}}) \exp\left[ -\frac{(1 - \tilde{t} - \tilde{z})^2}{4\tilde{\mathcal{D}}_{\text{eff},i} \tilde{t}} \right] d\tilde{z} \tag{15}$$

where $\tilde{\mathcal{D}}_{\text{eff},i}$ and $\tilde{t}$ are defined as in Eqs. (12) and (14).

Under the typical CPOT flow rate of 2 L min$^{-1}$ and vapor molecular diffusivity $1 \times 10^{-5}$ m$^2$ s$^{-1}$, $\tau_{c,\text{cyld}} = 1290$ s $\gg \frac{\tau_{c,\mathcal{D}_i}}{3.83^2} = 50$ s; therefore, the Taylor dispersion approximation for the gas-molecule RTD applies, and Taylor dispersion can be expected





to be important. Note that the presence of the static mixer and conical diffuser in the inlet section alters the input distribution of vapor molecules and particles at the entrance of the reaction section (Fig. 5C) from an idealized uniform initial condition, and Eq. (12) will not hold exactly for fitting of the results from actual pulse RTD experiments. The convolution (Eq. (15)) of the skewed input shape must be numerically calculated. The actual RTD of the reactor should also include the RTDs in the exit
cone and sample line.

### 4.3.2  Particle Residence Time Distribution

For the behavior of particles in the reactor, in general, the following processes need to be accounted for: (1) advection; (2) Brownian diffusion; (3) gravitational settling; and (4) growth/shrinkage owing to mass transfer from or to the gas phase. The particle number concentration distribution as a function of particle diameter $D_p$ is denoted $n(D_p, r, z)$. We do not include
particle-particle coagulation among the processes of importance, as, for typical number concentrations expected to be used, the timescale associated with coagulation will be long as compared to the typical residence time in the reactor (Seinfeld and Pandis, 2016). Gravitational settling of particles in a horizontal tubular flow reactor occurs as particles fall across streamlines and deposit on the lower half of the tube. (While cylindrical coordinates are usually employed in a flow tube reactor, it will prove to be advantageous to use a Cartesian coordinate framework for the numerical simulation of particle settling in horizontal
laminar flow in a tubular geometry.)

In general, particles undergo both Brownian diffusion in the flow as well as settling under the influence of gravity. Collectively, these processes give rise to particle loss by deposition on the wall during transit through a laminar flow tube reactor. The Brownian diffusion coefficient of a 80 nm diameter particle is approximately four orders of magnitude smaller than that of a typical vapor molecule. Consequently, for typical particle sizes and residence times in the reactor, the Brownian diffu-
sion of particles can be neglected, except for the region very close to the wall, wherein diffusive particle uptake at the wall can occur. To assess the effect of gravitational settling of particles, one needs to compare the characteristic settling distance during transit through the reactor, $v_s \tau_{c,\mathrm{cyld}}$, with the tube radius, $R$, where $v_s$ is the particle settling velocity. Figure 7 shows the size-dependent settling velocity and particle diffusivity for spherical particles. Under typical operating conditions, particles introduced uniformly across the entrance will tend to settle somewhat during transit down the reactor, so this process needs
to be accounted for in analyzing particle RTDs. The full equation describing the motion of particles in the horizontal tubular laminar flow under simultaneous diffusion and settling cannot be easily solved. As suggested by the particle-size dependence of settling velocity and diffusivity in Fig. 7, consideration of the two separate regimes, i.e., diffusion and settling, respectively, can simplify the problem.

For non-diffusive particles, particle motion in a horizontal tubular laminar flow is governed by the following differential
equations for particle position, $(x(t), y(t), z(t))$, in a Cartesian coordinate system (with origin at the center of the tube at $t = 0$, as shown in Fig. 8):

$$\frac{dx}{dt} = 0 \tag{16}$$



$$\frac{dy}{dt} = -v_y(D_p) = -v_s(D_p) \tag{17}$$

$$\frac{dz}{dt} = v_z(x,y) = U_{\max}\left(1 - \frac{x^2 + y^2}{R^2}\right) \tag{18}$$

Given an initial particle position, $x(0) = x_0$, $y(0) = y_0$, $z(0) = z_0$, this set of equations can be solved either numerically or analytically. Examples of the numerical simulation of particle trajectories are shown in Fig. 8. The analytical solution of Eqs. $(16) - (18)$ for the time $\tau$ that a particle resides in the flow is:

$$\tilde{t}^3 - 2\tilde{y}\tilde{t}^2 - (1 - \tilde{x}^2 - \tilde{y}^2)\tilde{t} + t_1/t_2 = 0 \tag{19}$$

where $\tilde{x} = x_0/R$, $\tilde{y} = y_0/R$, $t_1 = L_{\text{cyld}}/U_{\max}$, $t_2 = R/v_s$, and $\tilde{t} = t/t_2$. $\tilde{x}$ and $\tilde{y}$ are subject to the condition:

$$\tilde{y}(1 - \tilde{x}^2) - \frac{1}{3}\tilde{y}^3 - \frac{t_1}{t_2} + \frac{2}{3}(1 - \tilde{x}^2)^{3/2} \leq \tilde{x}^2 + \tilde{y}^2 \leq 1 \tag{20}$$

The integral over this closed space leads to the penetration efficiency $\eta$ for non-diffusive monodisperse particles:

$$\eta = \frac{2}{\pi}\left(-2\epsilon\sqrt{1 - \epsilon^{2/3}} + \epsilon^{1/3}\sqrt{1 - \epsilon^{2/3}} + \arcsin\sqrt{1 - \epsilon^{2/3}}\right) \tag{21}$$

where $\epsilon = \frac{3t_1}{4t_2}$ and the implicit condition here is that $\epsilon \leq 1$, i.e. $v_s \leq \frac{4R}{3L_{\text{cyld}}}U_{\max}$, otherwise $\eta = 0$. Calculated theoretical RTD and $\eta$ are shown in Fig. 9.

For diffusion-dominant particles (i.e., those of very small size), the settling velocity can be ignored, and the RTD of a pulse input can be approximated by the residence time along each streamline:

$$\langle n \rangle(t, L_{\text{cyld}}) = \begin{cases} 0 & 0 \leq t < t_1 \\ \frac{2N_0 t_1^2}{AU_{\text{avg}}t^3} & t \geq t_1 \end{cases} \tag{22}$$

Since actual particles undergo some degree of radial Brownian diffusion, which is not considered in Eq. (22), the full RTD should exhibit a broader and smoother profile than that predicted by Eq. (22) (as simulated by COMSOL, see Section 5.1.2).

Penetration efficiency ($\eta$) for diffusive mono-disperse particles can be calculated by Eqs. (8) and (9), where removal of particles at the wall is assumed. This is consistent with the boundary condition of the non-diffusive particles.

## 5   Results and Discussion

### 5.1   Experimental Evaluation of Penetration Efficiency and RTD

We present here the results of experimental evaluation of the RTD in the CPOT for both vapor molecules and particles. The
penetration efficiency ($\eta$) was determined by measuring a constant input of either gas-phase species ($SO_2$, $O_3$, and $H_2O_2$) or



polydisperse ammonium sulfate particles, immediately after the static mixer and at the exit. The RTD profiles were determined by introducing a 30 s pulse of $O_3$ or polydisperse ammonium sulfate particles into the reactor under dry conditions (RH < 1%). All experiments were performed at the typical operating flow rate of 2 L min$^{-1}$ in at least triplicate. The average residence time ($\tau_{avg}$) was obtained from each RTD profile according to:

$$\tau_{avg} = \frac{\Sigma I_j t_j}{\Sigma I_j},  \tag{23}$$

where $I_i$ is the signal recorded at each time step $t_j$.

### 5.1.1 Vapor molecules

The measured $\eta$ value of all the gas-phase species is essentially 100%. Each of $O_3$, $SO_2$, and $H_2O_2$ exhibited negligible interaction with the reactor wall; in future experiments, it is anticipated that organic vapors may behave differently. The extent of wall deposition of organic vapors in the flow tube reactor requires comprehensive study and will be addressed in a future publication.

A typical value of diffusivity, $1\times10^{-5}$ m$^2$ s$^{-1}$, is used in COMSOL to predict the gas-phase RTD. The measured and predicted gas-phase RTD are shown in Fig. 10A. A large discrepancy was observed between the measured and theoretical RTD under presumed isothermal conditions. The predicted gas-phase RTD exhibits a symmetrical distribution centered at approximately 27 min. However, the measured RTD of gas-phase $O_3$ exhibits an asymmetrical feature, somewhat similar to the particle RTD (Fig. 10B). The $\tau_{avg}$ values obtained from the $O_3$ pulse experiments and simulations are also summarized in Fig. 10A. The measured $\tau_{avg}$ value of $O_3$ is shorter than predicted by 1.5 min. Potential explanations for measured RTDs are discussed in Section 5.2.

### 5.1.2 Particles

Figure 11 shows the measured size distributions at the inlet and outlet, as well as the size-dependent penetration efficiency obtained as the ratio between the two. The theoretical particle $\eta$ curves under the influence of loss by gravitational settling and diffusion have also been calculated by applying the CPOT parameters to Eqs. (9) and (21) (Fig. 11). Only the reaction sections of the CPOT were considered in this theoretical calculation (i.e. the inlet and exit cones are excluded). We consider this calculation as a qualitative guideline for $\eta$ in the CPOT. Settling velocity and diffusivity of particles are size-dependent (Fig. 7), resulting in reduced transmission for very small and large particles, due to diffusion loss and gravitational settling, respectively. Both measurements and theory indicate that $\eta$ is maximized at a particle diameter of approximately 100 nm. The measured maximum penetration efficiency is $\sim$ 80 %, indicating a loss of particles. Note that since we measured the particle size distribution after the static mixer, this loss does not arise from the static mixer. The behavior of particles in the exit cone is difficult to predict and may reflect a certain extent of particle loss. Another possible explanation for the RTD discrepancy is the increasingly significant electrostatic loss to the wall caused by evaporation of particle-borne water. Although the atomized particles were introduced through a diffusion drier, the CPOT is operated under a very dry condition. Additional evaporation





of particulate water (at different evaporation rates) may have led to reduced particulate mass and enhanced electrostatic force on the particles, possibly contributing to the particle loss in the CPOT not accounted for in the theoretical simulation.

A typical value of particle diffusivity, $1 \times 10^{-9}$ m$^2$ s$^{-1}$, is used in COMSOL to predict the RTD. Figure 10B compares the measured RTD of polydisperse ammonium sulfate particles to that of the COMSOL simulation. Under isothermal conditions, the particle RTD exhibits a zigzag feature which is likely due to the static mixer that may distribute particles somewhat unevenly across streamlines, as can be seen in the velocity profile in Fig. 5A. As laminar flow develops in the reaction section, particles follow their respective streamlines until the exit cone, appearing as the zigzag pattern on the RTD profile. This zigzag feature was absent in the vapor molecule RTD, likely due to the larger diffusivity of vapor molecules. The theoretical RTD of particles in an idealized laminar flow reactor exhibits a sharp peak when the center line first arrives at the exit (Eq. 22). The experimental RTD observed exhibits a rather gradual rise likely due to the method of introduction instead of a sharp pulse (Fig. 5C). Figure 10B shows that, under isothermal conditions, the modeled RTD reproduces the shape and the peak time of the observed RTD, and the $\tau_{avg}$ values also shows excellent agreement. However, the modeled RTD appears narrower than the experimental one. This indicates that particles arrive earlier and remain for a longer time than COMSOL predicts. This discrepancy is the result of some degree of non-ideal flow in the reactor.

## 5.2 Non-ideal Flow in the Reactor

### 5.2.1 Non-isothermal Effects

The discrepancy between isothermal laminar flow theory and the experimental results can be attributed in part to non-isothermal conditions in the CPOT. As noted earlier, the Richardson number (Eq. (1)) criterion indicates that a minute temperature difference (0.007 K) between the bulk and the wall can induce recirculation flows. The measured particle RTDs under isothermal conditions are compared to that obtained under maximum UVA radiation in Fig. 10B. A pronounced difference is that the RTD curve under radiation appears much smoother. The $\tau_{avg}$ value under irradiation is shortened by 1.5 min compared to that under isothermal conditions. Given the close agreement between the two RTD profiles, it is unlikely that a recirculation within the tube has been established. It is more likely that the slight non-isothermal condition has created secondary flows that act to mix the tracers both radially and axially; we discuss this hypothesis in Section 5.2.2.

To further investigate non-isothermal effects, the temperature of the water jacket was raised in a step-wise manner to approach a significant temperature difference between the bulk flow and the wall. The experiments were conducted in the absence of UV radiation. The injected air was at room temperature (approximately 23°C), so a higher water jacket temperature is expected to exacerbate the deviation from isothermal conditions. The results of these experiments are shown in Fig. 12. The RTD at each temperature is the average of 3 to 4 replicates. As shown in Fig. 12A, the RTD at 25°C appears indistinguishable from that at quasi-isothermal conditions (the dashed line, we use "quasi-isothermal" here to distinguish from strict isothermal conditions in the model). Particles arrive at the exit cone earlier at higher water jacket temperatures, mirroring the observed discrepancy between the modeled and observed RTD profiles. This trend is clearly illustrated by Fig. 12B, where the arrival





time of particles in each experiment is shown as a function of the water jacket temperature. This observation is consistent with the hypothesis that a difference in temperature between the wall and the inlet flow leads to the non-ideal conditions.

### 5.2.2  Potential Presence of Secondary Flows in the CPOT

The model-experiment deviation in RTD profiles has likely arisen from secondary flows induced by the the non-isothermal

condition. A idealized model of the temperature effect on the recirculation in the CPOT was given in Section 3.3. Consider that the wall of the reactor is at a constant room temperature as slightly cooler air is introduced into the reactor. Two orthogonal forces interact with each other in the horizontal flow tube when they are of similar orders of magnitude: forced convection by the pressure gradient (horizontal) and buoyancy-induced free convection (vertical). The actual velocity field in this situation is difficult to simulate (Iqbal and Stachiewicz, 1966; Mori and Futagami, 1967; Faris and Viskanta, 1969; Siegwarth et al.,

1969). Generally, to satisfy mass conservation, the air close to the wall is warmed and rises along the side wall, inducing a downward flow in the center of the tube, forming two symmetric vortices. Superposition of the primary forced convective and the secondary free convective flows convert the vertical recirculation into spiral motions along the tube. The spiral flow developed in the reaction section plays a similar role as the static mixer in the inlet section. To quantitatively represent this effect, one can introduce an enhanced isotropic eddy-like diffusivity ($\mathscr{D}_e$), a statistical fluid field related property that can better

mix vapor molecules and particles than their inherent diffusivity.

To verify the presence of the spiral secondary flow in the CPOT, we have systematically increased the diffusivity used in the COMSOL simulations. The agreement between simulated and observed RTD improves as the value of $\mathscr{D}_e$ is increased in the COMSOL simulation, with the optimal agreement being achieved when $\mathscr{D}_e$ is set at $4.5\times10^{-4}$ m$^2$ s$^{-1}$ and $6.0\times10^{-4}$ m$^2$ s$^{-1}$ for O$_3$ and particles, respectively (Fig. 13). These $\mathscr{D}_e$ values are, respectively, 45 and $6\times10^5$ times the diffusivity of

vapor molecules and particles from the strictly parabolic flow base case (Fig. 10). The vapor molecule RTD (Fig. 13A) no longer exhibits the symmetrical feature of the base case, due to the enhanced Taylor dispersion. The particle RTD (Fig. 13B) is also substantially broadened compared to the base case and exhibits close agreement with the observations. The optimal $\mathscr{D}_e$ values for vapor molecules and particles are similar, suggesting that the molecular diffusion in the CPOT is dominated by the secondary flows. This offers an explanation for the similarity in the observed RTD profiles of O$_3$ and particles, despite orders

of difference in their inherent diffusivity.

To evaluate further the $\mathscr{D}_e$ values determined from the COMSOL simulations and the hypothesis of secondary flows, one can adopt a separate approach to examine $\mathscr{D}_e$. Given the mixing provided by the static mixer and the conical diffuser, the optimal values of $\mathscr{D}_e$ can be applied in Eq. (14). The values of $U_{avg,fit}$ and $\tau_{c,cyld,fit}$ are adjusted to find the best match between Eq. (14) and the observed RTD profiles. The optimal fitting results are shown in Fig. 13. The fitted average flow velocity ($U_{avg,fit}$) is

$2.1\times10^{-3}$ m s$^{-1}$, which results in a characteristic residence time $\tau_{c,cyld,fit}$ of 1360 s. This $U_{avg,fit}$ value agrees well with the designed average velocity ($2.0\times10^{-3}$ m s$^{-1}$). Applying other $\mathscr{D}_e$ in Eq. (14) resulted in a deviation of $U_{avg,fit}$ from the designed value. This observation again suggests that the non-isothermal secondary flow induced eddy-like diffusion dominates the mass transport process in the CPOT.



Overall, these results highlight the importance of temperature effects in approaching an ideal flow condition in a laminar flow reactor. Even a small temperature deviation can likely create secondary flows in the flow field that affect both the RTD and the $\tau_{avg}$ of tracers. It is to be noted that these secondary flows occurring at Re $\cong$ 20 have no relationship with classic turbulent flow.

## 6  Conclusions

A laminar flow tube reactor for atmospheric chemistry studies, the Caltech PhotoOxidation flow Tube reactor (CPOT), has been designed with the goal of achieving a well-defined fluid environment to determine accurately reaction conditions. By characterizing the fluid dynamics in the CPOT, and also exploring fundamental concepts governing the behavior of gas and particles in a flow tube reactor, the current study reveals a number of implications for design and implementation of laminar
flow tube reactors.

Inlet design plays a significant role in establishing the fluid dynamic environment in a flow tube reactor. The CPOT inlet comprises a conical diffuser following an upstream static mixer. Computational fluid dynamics (CFD) simulations demonstrate that this injection scheme introduces flow into the reaction sections avoiding flow separation from the wall, assisting a rapid transition to a parabolic profile under idealized, isothermal conditions.

The actual fluid dynamics within the CPOT was examined experimentally by comparing the penetration efficiency ($\eta$) and residence time distribution (RTD) of particles and vapor to those predicted under ideal flow conditions. Vapor molecules examined in the current study ($O_3$, $H_2O_2$ and $SO_2$) exhibited 100 % transmission, indicating no removal on the walls of the reactor. The penetration efficiency of polydisperse ammonium sulfate particles was calculated by considering two separate regimes, one dominated by diffusion loss and the other by gravitational settling. The penetration efficiency calculated with this
approach reproduced the observation satisfactorily.

Comparison of the modeled and observed RTD emphasizes the importance of temperature control in approaching ideal laminar flow in a flow tube reactor. Despite the temperature-controlled water jacket, model-experimental discrepancy in the RTD profiles was likely attributed to slightly non-isothermal conditions in the CPOT, as the discrepancy increased as the system became more and more non-isothermal. Thus, non-isothermal conditions in the CPOT have likely created the secondary flows
that give rise to the difference in the modeled and observed RTD profiles. This conclusion was supported by substantially improved agreement between the modeled and observed RTD when an enhanced eddy-like diffusivity ($\mathcal{D}_e$), considerably larger than the inherent diffusivity of vapor molecules and particles, was employed in the CFD simulations. The optimal $\mathcal{D}_e$ values obtained in the current study ($4.5\times10^{-4}$ m$^2$ s$^{-1}$ and $6.0\times10^{-4}$ m$^2$ s$^{-1}$ for $O_3$ and particles) were sufficiently large that the mass transfer of tracers in the CPOT is likely dominated by the secondary flows. More generally, the current study
indicates that these secondary flows can exist in laminar flow tube reactors and can significantly affect the fluid dynamics and the reaction conditions.

In spite of these non-idealities, the simulations and experimental studies demonstrate that the combination of the conical diffuser with entrance vorticity arising from the static mixers allows a large laminar flow reactor to approach the parabolic



flow that is needed for quantitative kinetic studies. The measured residence time distribution will enable correction for the unavoidable deviations that do occur. Finally, the perturbations from strict laminar flow in the horizontal tube are a result of buoyancy effects. If it had been feasible to mount the flow tube vertically, these effects could have been largely eliminated.

*Acknowledgements.* We gratefully acknowledge a generous gift by Christine and Dwight Landis to support the construction of this reactor.
5   We also thank Professor Paul Wennberg for useful discussions and for offering laboratory supplies. This work was supported by National Science Foundation grant AGS-1523500. Ran Zhao was supported by the Natural Science and Engineering Research Council of Canada.





## Appendix A:  List of Symbols

| Symbol | Meaning | unit |
|---|---|---|
| $A$ | cross section of the reactor | $\text{m}^2$ |
| $\bar{c}$ | mean molecular speed | $\text{m s}^{-1}$ |
| $c$ | concentration | $\text{mol m}^{-3}$ |
| $D$ | reactor diameter | m |
| $D_p$ | particle diameter | nm |
| $\mathscr{D}$ | diffusivity | $\text{m}^2\ \text{s}^{-1}$ |
| $g$ | gravitational acceleration | $\text{m s}^{-2}$ |
| $k_w$ | mass transport coefficient to the wall | $\text{m s}^{-1}$ |
| $L$ | length of the reactor | m |
| $n$ | particle number concentration | $\text{cm}^{-3}$ |
| $N_0$ | total moles or number of the pulse input | mole or number |
| $P$ | generation rate of species | $\text{molec cm}^{-3}\ \text{s}^{-1}$ |
| $R$ | radius of the reactor | m |
| $S$ | consumption rate of species | $\text{molec cm}^{-3}\ \text{s}^{-1}$ |
| $t_0$ | duration | s |
| $T$ | temperature | K |
| $U$ | characteristic velocity of the fluid | $\text{m s}^{-1}$ |
| $v$ | velocity | $\text{m s}^{-1}$ |
| $V$ | volume of the reactor | $\text{m}^3$ |
| $Q$ | volumetric flow rate | $\text{m}^3\ \text{s}^{-1}$ |
| ***Greek*** | | |
| $\beta$ | thermal expansion coefficient of fluid | $\text{K}^{-1}$ |
| $\mu$ | viscosity of the fluid | $\text{kg m}^{-1}\ \text{s}^{-1}$ |
| $\nu$ | kinematic viscosity | $\text{m}^2\ \text{s}^{-1}$ |
| $\rho$ | density of fluid | $\text{kg m}^{-3}$ |
| $\theta$ | angle of the cone | $^\circ$ |
| $\tau$ | residence time | s |



## Appendix B: List of Dimensionless numbers and subscripts

| Symbol | Name | Expression |
|---|---|---|
| *Greek* | | |
| $\gamma$ | uptake coefficient | |
| $\eta$ | penetration efficiency | |
| $\epsilon$ | ratio of time scale of convection to that of settling | $3L_{\text{cyld}}v_s/4RU_{\text{max}}$ |
| $\xi$ | dimensionless length | $\pi\mathscr{D}L_{\text{cyld}}/Q$ |
| *Dimensionless groups* | | |
| Gr | Grashof number | $g\beta D^3\Delta T/\nu^2$ |
| Pe | Péclet number | $2RU_{\text{avg}}/\mathscr{D}_i$ |
| Re | Reynolds number | $\rho U_{\text{avg}}D/\mu$ |
| Ri | Richardson number | Gr / Re$^2$ |
| *Subscripts* | | |
| c | characteristic value | |
| cyld | cylindrical tube | |
| e | eddy-like | |
| fit | fitted result | |
| $i$ | species | |
| $j$ | time step | |
| s | settling | |
| avg | average value | |
| eff | effective value | |
| entr | entrance | |
| in | inlet | |
| max | maximum value | |
| out | outlet | |
| r | r-component in cylindrical framework | |
| x | x-component in Cartesian framework | |
| y | y-component in Cartesian framework | |
| z | z-component in Cartesian or cylindrical framework | |
| *Supscript* | | |
| ~ | nondimensionalized variable | |



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


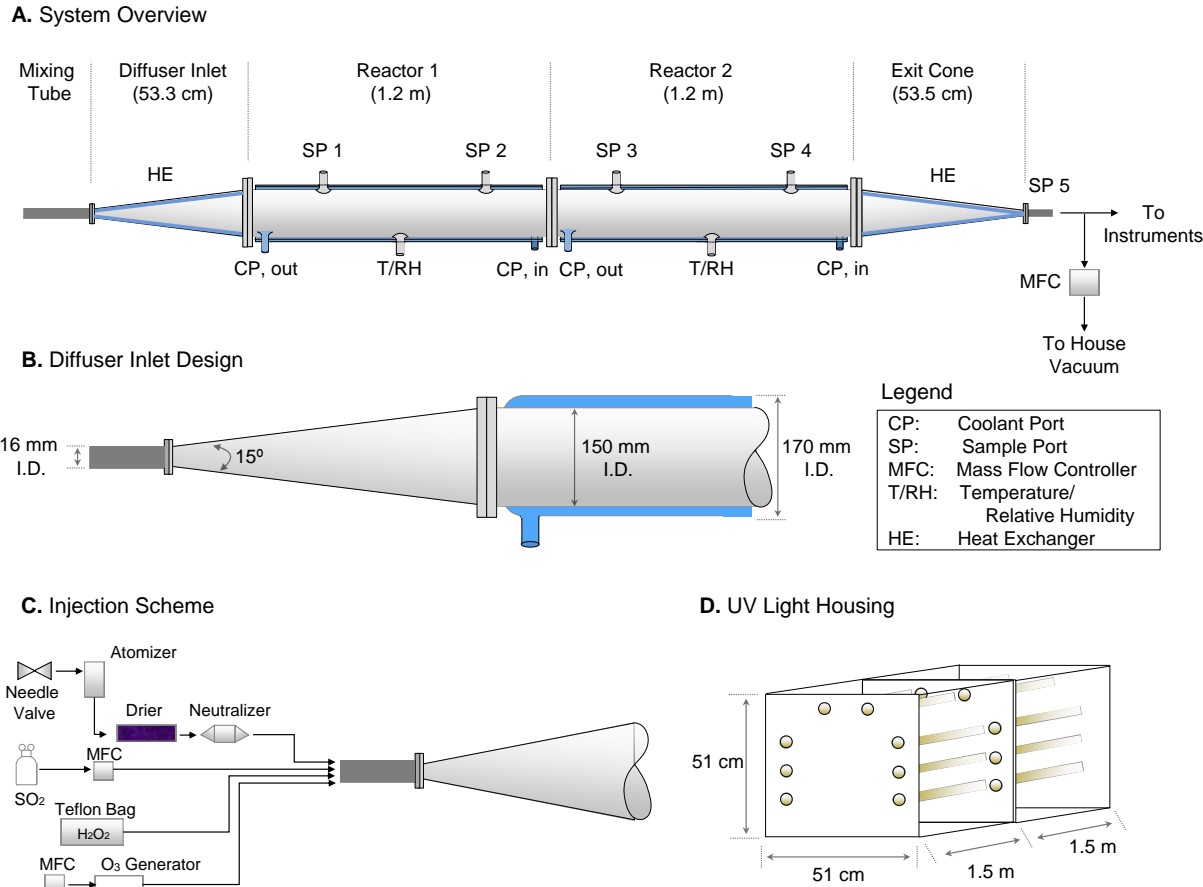

**Figure 1.** Overall schematic of the Caltech PhotoOxidation flow Tube (CPOT). (A) The inlet design. (B) The injection scheme. (C) Schematic for the housing chamber and UV lamps (D).





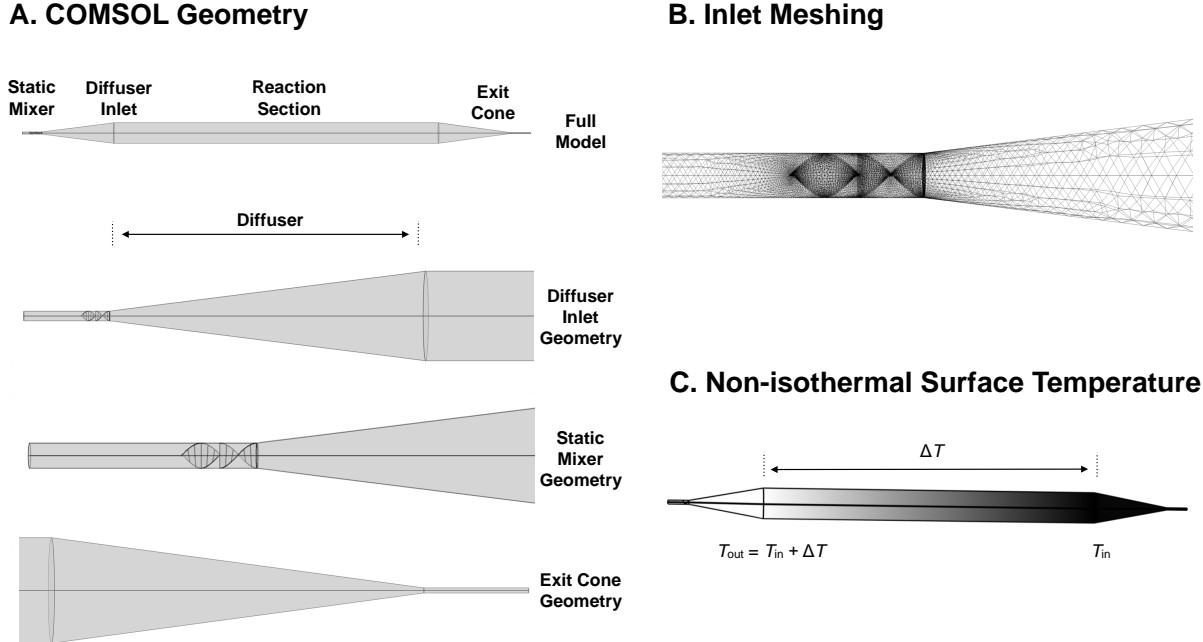

**Figure 2.** (A) Overview of the geometry used to simulate flow and species transport within the CPOT. (B) Inlet meshing for static mixer. (C) Schematic illustration of the temperature gradient used in non-isothermal simulations.





## A. Straight Tube Inlet

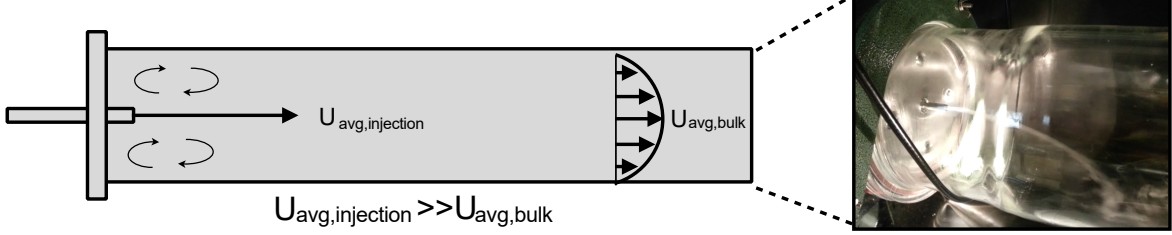

## B. Showerhead Inlet

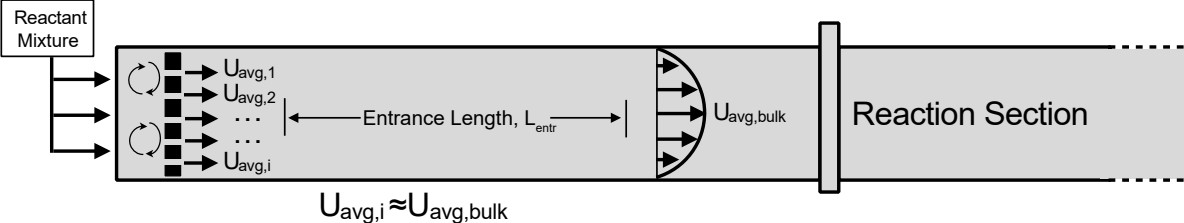

## C. Diffuser Inlet

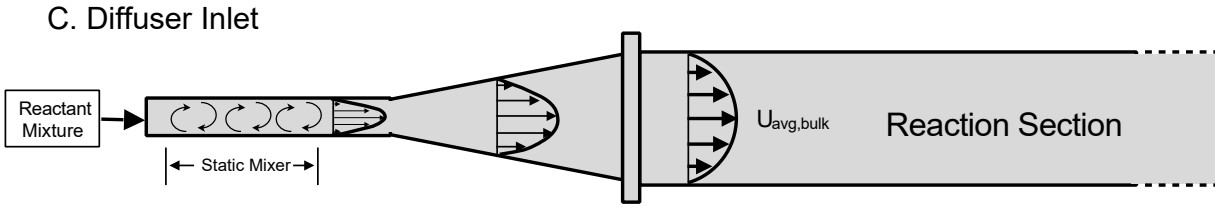

**Figure 3.** Inlet designs exemplified by (A) the Potential Aerosol Mass reactor (PAM, Kang et al., 2007; Lambe et al., 2011a) (B) the UC Irvine flow tube (Ezell et al., 2010) and (C) the CPOT. $U_{avg,injection}$, $U_{avg,bulk}$, and $U_{avg,i}$ values denote the average velocities at the PAM inlet, in the bulk reaction section, and at the exit of a showerhead hole, respectively. Inlet (A) also illustrates the "fire hose" effect, as demonstrated by the injection of smoke in a Pyrex glass tube.





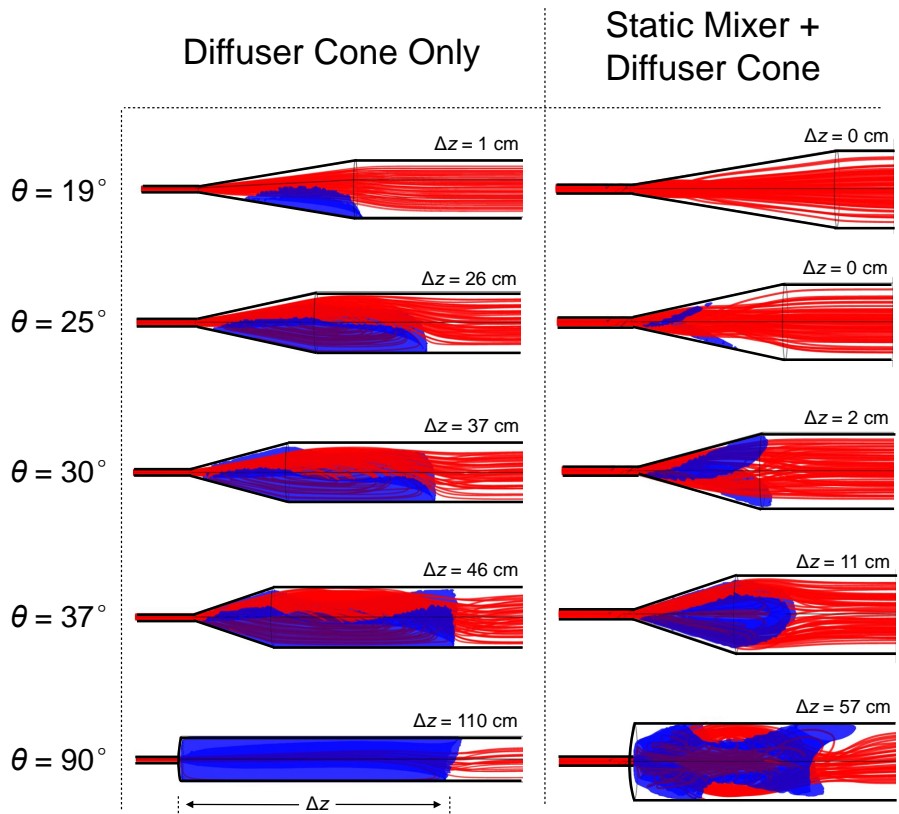

**Figure 4.** COMSOL simulation results for a suite of diffuser angles assuming isothermal conditions. Images in the left column are from simulations conducted in the absence of a static mixer. Images in the right column are from simulations employing a 2-element static mixer upstream of the diffuser cone. The red traces are streamlines demonstrating the flow pattern of fluid introduced upstream of the static mixer. The blue surfaces illustrate regions where the axial velocity $< 0$ m s$^{-1}$. Together, these traces illustrate the recirculation zone. $\Delta z$ is the length that the recirculation zone penetrates into the reaction section. All simulations were performed for a volumetric reactor flow of 2 L min$^{-1}$.





**Figure 5.** COMSOL simulated velocity field at the inlet of the CPOT under isothermal conditions. Simulations were performed for the actual CPOT design: a 15° diffuser cone coupled to a static mixer. The velocity magnitude at various axial positions is shown in (A), and 1-D axial velocity profiles within the "inlet-affected" region are shown in (B). The velocity magnitude in (B) is presented on the axis below each velocity profile. Note that DL = Diffuser Length (53.3 cm). (C) shows the normalized residence time of vapor molecules and monodisperse particles at various axial positions. A 30 s square wave pulse is used as the input.





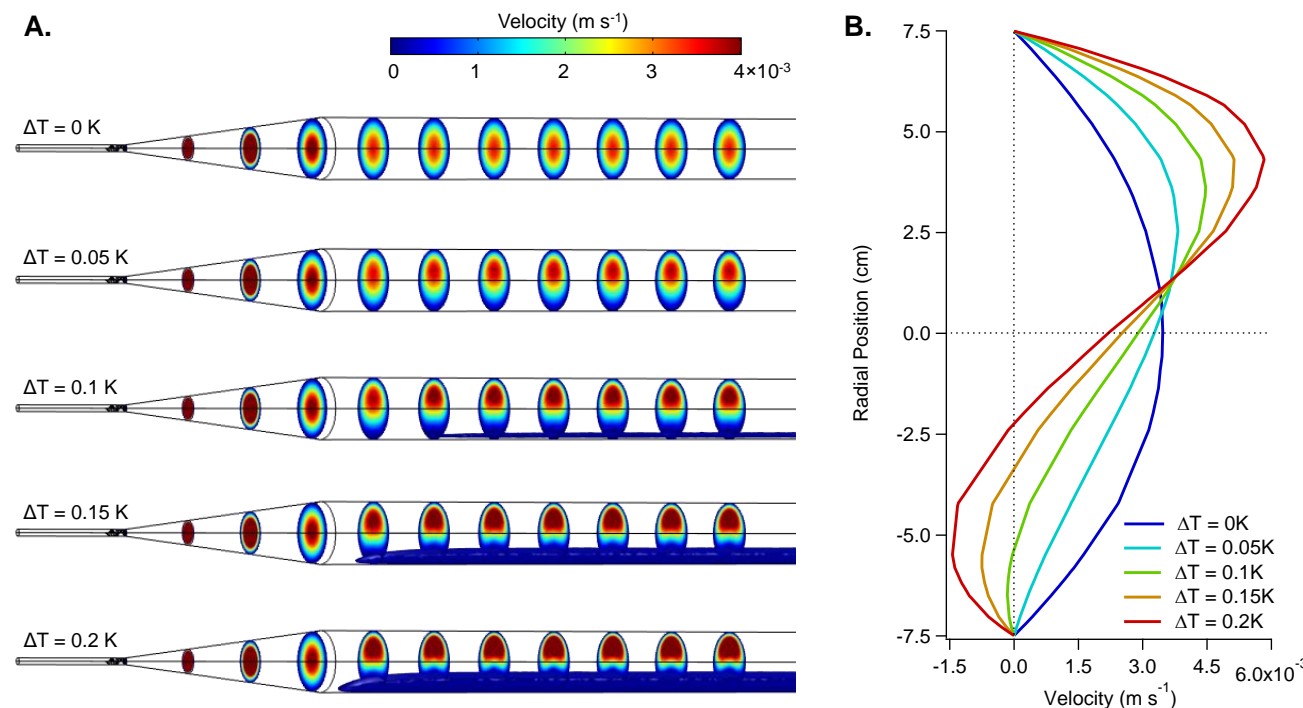

**Figure 6.** COMSOL simulated flow profiles as a function of reactor wall temperature gradient $\Delta T$. (A) 3D simulation results demonstrating cross-sectional velocity profiles and consequential recirculation zones (blue isosurface). (B) 1D velocity profiles at axial position $z = 150$ cm.



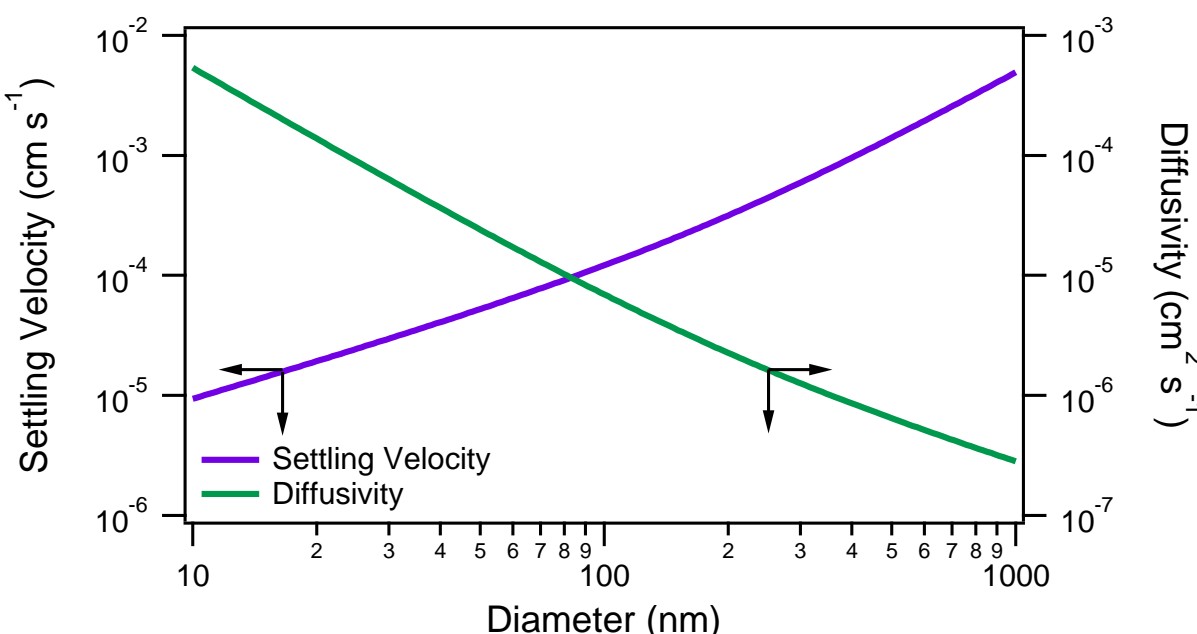

**Figure 7.** Particle settling velocity and Brownian diffusivity for spherical particle of unit density as a function of particle diameter (Seinfeld and Pandis, 2016).




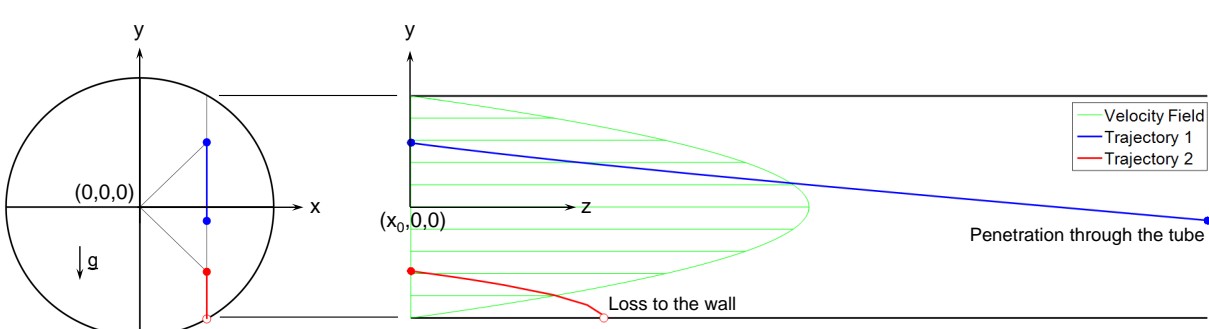

**Figure 8.** Particle trajectories in a vertical plane in a laminar flow tube. Particles are of the same size and are subject to gravitational settling and fluid advection. The Cartesian coordinate framework is indicated. Two different cases are shown: blue particles are those that can successfully penetrate through the tube, while red particles eventually deposit on the tube wall.





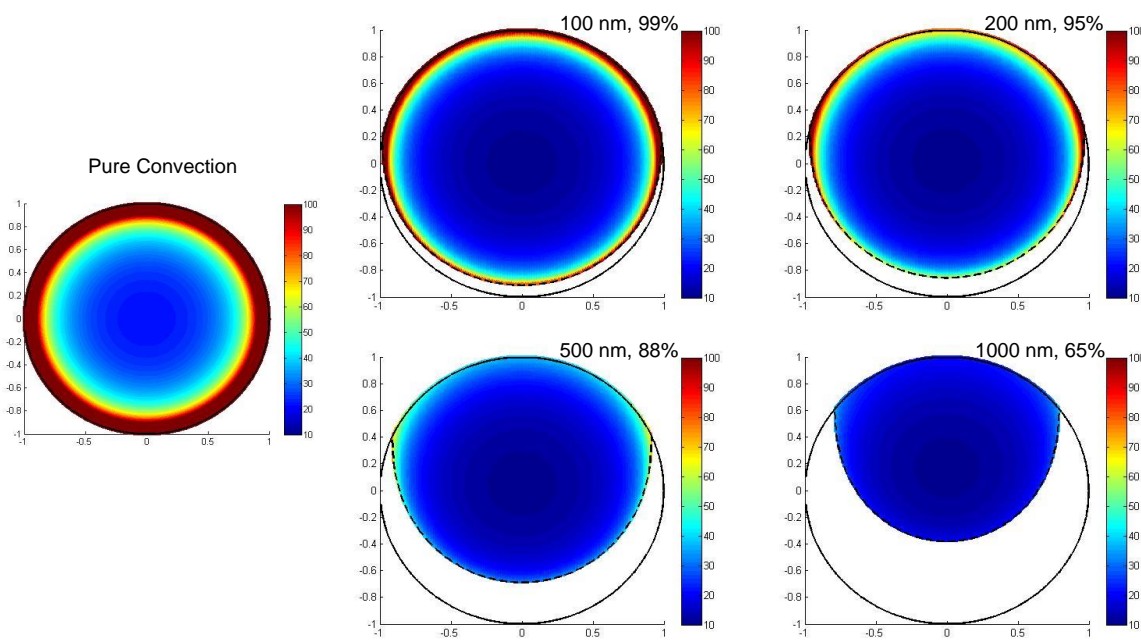

**Figure 9.** Theoretical residence time distribution (Eq. (19)) and penetration efficiency (Eq. (21)) of particles with different diameters in a horizontal flow tube. Only gravitational settling and convection are considered. The simulation assumes a uniform distribution of monodisperse particles at the entrance of a well-developed laminar flow with no interaction between particles. Each point corresponds to the residence time and the initial position of the particle. The color bar indicates the residence time (min). The open space between the dashed curve and the tube wall indicates the region in which particles have deposited on the tube wall. The residence time of purely convective, non-diffusion particles (Eq. (22)) is shown for reference.





**Figure 10.** Experimental and COMSOL simulated residence time distributions of (A) $O_3$ vapor molecules and (B) polydisperse ammonium sulfate particles. The diffusivity used in COMSOL for $O_3$ is $1 \times 10^{-5}$ m$^2$ s$^{-1}$ and for particles is $1 \times 10^{-9}$ m$^2$ s$^{-1}$. The average residence time in each case is compared in the insets as reference.

.




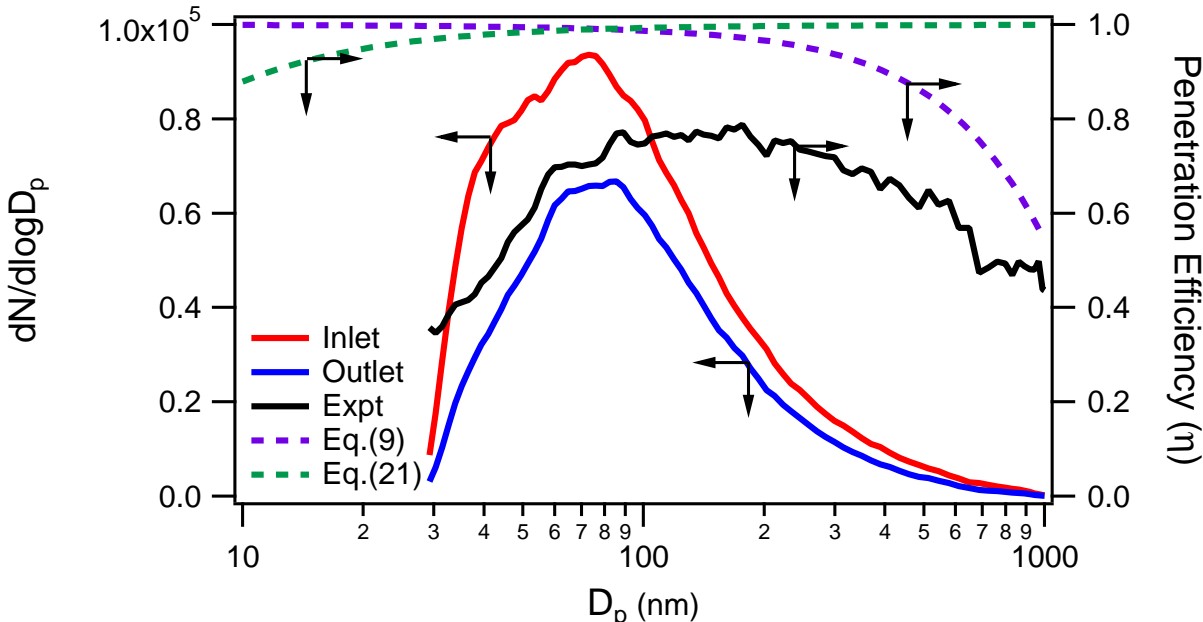

**Figure 11.** Measured ammonium sulfate particle size distributions at the inlet and outlet of the CPOT, as well as the penetration efficiency derived from these measurements. The calculated penetration efficiency with respect to particle diffusion loss (Eq. (9)) and gravitational settling (Eq. (21)) are indicated by the dashed lines.




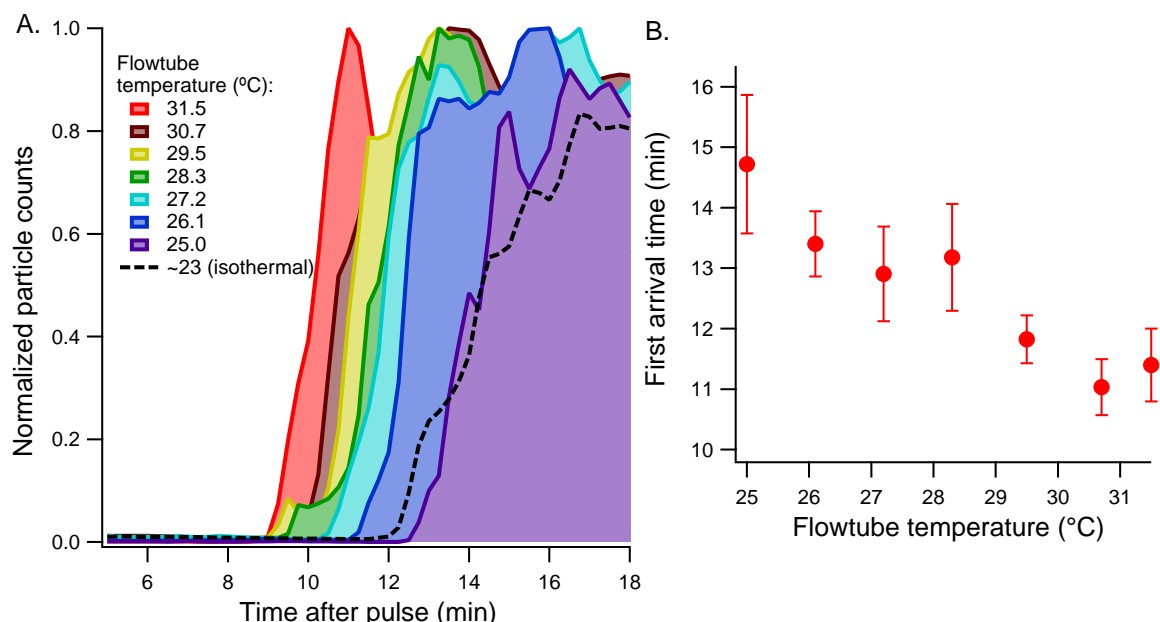

**Figure 12.** Non-isothermal effects on particle RTD. The water jacket temperature was raised systematically against the room temperature (∼23 °C). (A) Normalized ammonium sulfate particle counts recorded at the exit of the CPOT are plotted against time after a pulse is introduced at the inlet. The experiment at each temperature is repeated 3 to 4 times. The results from a set of isothermal experiments are also included (dashed line) for reference. (B) Arrival times of the first major peak of each experiment.

.





**Figure 13.** Comparison of experimentally determined RTD of (A) $O_3$ vapor molecules and (B) polydisperse ammonium sulfate particles to optimized simulation results employing an eddy-like diffusivity ($\mathscr{D}_e$). The COMSOL simulation employs an optimal $\mathscr{D}_e$ values of $4.5 \times 10^{-4}$ $m^2$ $s^{-1}$ for $O_3$ and $6.0 \times 10^{-4}$ $m^2$ $s^{-1}$ for particles. The fittings of Eq. (14) employs the same optimal $\mathscr{D}_e$, as well as an optimal average velocity ($U_{avg,fit}$) of $2.1 \times 10^{-3}$ m $s^{-1}$ and an optimal characteristic residence time ($\tau_{c,cyld,fit}$) of 1360 s.