# Peer review of "The Caltech Photooxidation Flow Tube Reactor - I: Design and Fluid Dynamics"

_Atmospheric Measurement Techniques, 2016_

## Referee Comment (RC1) · Anonymous Referee #1 · 5 Oct 2016

Review of Huang et al., AMT

General comments:

Huang et al. presents a nice manuscript about the new CalTech Photooxidation Flow Tube Reactor (CPOT), which will enable them to join the dozens of research groups who have been using Oxidation Flow Reactors to study the secondary particles and their properties for oxidation times scales appropriate for the atmosphere for a decade. The manuscript is well written and well organized. It contains lots of nice color figures and drawings and solid analyses. However, there are some errors and omissions which must first be addressed, but when they are, the manuscript will be suitable for AMT.

Specific comments:

[Figure]

When designing a flow reactor or chamber of any sort, the designer needs to consider how the flow reactor or chamber will be used. There are always compromises because there is no perfect flow reactor or chamber, particularly for atmospheric pressure. For instance, if you want to relate time in the reaction to reaction time, then you will have enhanced wall interactions because that is how this characteristic is achieved. For a flow tube that is at atmospheric pressure, you must have something to mix the different gases and particles to be studied completely into the carrier air flow. This mixing may be by shower heads or static mixers with lots of surface area, but then you will also have lots of possible surface interactions and losses for gases and particles before they even enter the main flow reactor.

If you want to study ambient aerosol particle chemistry or particle mass yields, then you will want to minimize wall effects and losses, which means minimizing wall interactions, but then the ability to relate time in the flow tube to reaction time may become difficult. If you want only to do laboratory studies, you still need to decide whether you want to look at chemical evolution (so that time in the reactor equals reaction time) or chemical properties that are distorted by wall interactions, such as SOA yields. Then that decision determines if you embrace wall interactions to give you laminar flow or you try to avoid walls and lose the ability to relate time in the chamber to reaction time. Which way the compromises are made depends entirely on the purpose. Of course you can decide to try to do both, and possibly compromise both.

The authors of this manuscript should have an introductory paragraph stating what they intend to study with this CPOT so that the reader can judge whether they have designed it appropriately for that purpose. They should also lay out the strengths and weaknesses of their approach for each type of study they intend to do (e.g., SOA mass yields, SOA chemical properties, time-dependent kinetic studies).

2.5 Computational Fluid Dynamics (CFD) Simulations page 4, line 20. Assuming isothermal flow is unrealistic, which the authors say later in the manuscript. The authors should say up front that they are going to try isothermal to see what they get and

then will need to modify this assumption later in the manuscript.

3.1 Injection method. Page 5, line 8. The authors are wrong about the method of injecting air into the PAM chamber. The flow does enter through a tube in the middle, but is quickly diverted by a covered cap with holes drilled sideways to divert the flow toward the outer edges of the chamber. This air is then passed through a fine-mesh screen to break the larger eddies into smaller eddies inside the actual chamber. I suggest that the authors get in touch with the builders of the PAM chamber, so that they know and are thus able to give a correct description of the actual flow in the PAM chamber.

Page 5, line 16. This is not the dead volume to which Lambe et al., are referring. I recommend that the authors contact the designers of the PAM chamber to make sure they understand exactly what the PAM chamber design is, as well as the compromises that were made and why.

Page 5, line 18. The PAM chamber used in Ortega et al., is basically the same as the one used in Lambe et al.. The only difference is that the entrance plate with the flow diverter is removed so as to completely eliminate the loss of ambient particles and gases on the entrance plate and diverter by having the air flow into the chamber over almost its entire cross section. It is not to "reduce recirculation" in the reactor, as stated in this manuscript, although it might have improved the flow somewhat. The main result was that there was no longer any ambient particle loss in the PAM chamber.

Section 3.2. Page 6, Can the light from the reactor section get into the diffuser section? Is there ozone in the diffuser section? If so, then shouldn't the authors consider the diffuser section part of the reactor?

How does the lack of laminar flow in initial part of the diffuser section affect the particle formation/chemistry that the author may want to study since the diffuser is part of the reactor?

Section 3.3. The authors did look at issues of convection due to the temperature gradients in CPOT, but not the convection due to "hot spots" along the walls, which are likely to be important, irregular, and possibly transient.

Did the authors measure the temperature distribution of the walls in the CPOT to within 0.1 C? I would assume that there will be hot spots with differences of as much as 0.2 C or more, even with the temperature-control jackets and especially when the lights are on. My guess is that assuming an axial temperature gradient in the non-isothermal case is the best case scenario and that reality will be quite a bit worse.

We could do the following analysis with the Richardson number, but we can illustrate the problem with an even simpler calculation. Simple buoyancy calculations would suggest that air warmed by these 0.2 C hot spots would stir the reactor vertically from bottom to top in less than 10 seconds, which is a much shorter time than the ~20-minute residence time. And, because the reactor is a round tube, this vertical motion will get translated into horizontal motion as the rising air hits the top half of the tube. The stirring due to these hots spots is likely to be as important as any other flow considerations and is probably a major driving force in their eddy diffusion correction. Such eddy diffusion will do a great job of bringing the air in the chamber into close contact with the walls several times, thus increasing wall loss and exchange.

The temperature gradients that the authors determine are important are about the same as the 0.2 C difference used in this simple calculation. Thus, they have every right to be worried about thermal gradients and should include hot spots in their thinking about how convection is going to be a huge problem.

5. Results and Discussion 5.1 Measuring the initial value of the gases and particles downstream of the static mixer does not give a complete picture of particle and gas loss if those particles and gases will be added upstream of the static mixer during the experiments. How much of the gases and particles is lost on the static mixer? My guess is that a significant amount of the SO2 and the ammonium sulfate particles will

be lost, as will some of the H2O2. Now, if the authors intend to measure the amounts of gases and particles downstream of the static mixer and use these values as their initial values, then this test would be valid. Is that their intent?

Also, what about any gases / particles that are modified by interactions with the static mixer? They could be important for the chemistry that occurs in the CPOT reaction section.

Even for the way this penetration study was done, my guess is that the authors had to wait a while for the surfaces to acclimate to SO2 before they were able to get it all through. Did the authors have to wait?

Page14, line 8. This test does not provide evidence that there was negligible interaction with the walls for the gases. All the authors can really say is that there was negligible loss, which could mean that the surfaces were (temporarily?) passivated. How that would change with the lamps on and active chemistry is anyone's guess.

It would be good to know the penetration efficiency for both initial cases – before the static mixer and just after the static mixer. The authors should measure it and report the results in the next version of the manuscript.

Also, were these penetration efficiency tests done under dry conditions? If so, then these are "best case" values, especially for SO2 and H2O2. Unless the authors intend to do all their experiments under dry conditions (which will greatly limit the maximum OH exposure in CPOT and the applicability of the results to the real atmosphere), they should do the penetration efficiency tests under the same conditions as they will use in the experiments. My guess is that the penetration efficiency for SO2 and H2O2 will fall rather precipitously.

5.1.2. page 14, line 27. It is doubtful that the 20% particle loss is due to electrostatic loss to the quartz walls. Instead, it is likely that the loss is due to more active convection taking the particles closer to the wall more frequently in the CPOT than the authors

think.

Page 16, line 19. Why are the eddy diffusivities for ozone and particles different? I would think that they should be the same for small particles.

Somewhere at the end of the discussion and results, the authors need to compare the characteristics of the CPOT to those of other flow tube reactors, such as the PAM chamber, TPOT, and the Irvine chamber. From the data provided here, CPOT appears to be not much better than any of these other existing chambers in terms of particle transmission or residence time distribution. If you look at the difference in the actual arrival time to that predicted by laminar flow and the full-width and also at the half-maximum of the actual peak to that of the laminar flow case, as in Figure 10, and compare those to what is achieved in TPOT and PAM in Lambe et al. Figure 3, you will see that the actual arrival time is about half that for the laminar case and the FWHM peak width is about double that of the laminar case for all three flow tubes.

Therefore, it is unclear to me if the authors have achieved their goal of having a flow tube that can be used for kinetic studies (in the sense of time in the chamber is equal to reaction time) any better than the existing flow reactors can.

Conclusions. The authors say in the conclusion that "The measured residence time distribution will enable correction for the unavoidable deviations that do occur." The only way this statement can be made credible is for the authors to add a section that shows how they are going to do this and then demonstrate that it works with real kinetics measurements. Then they need to do an error analysis to show what the uncertainty of their kinetic measurements will be. If they do not wish to do this work for this manuscript, then I suggest that they remove this sentence.

Conclusions. The second-to-last sentence in the manuscript says "Finally, the perturbations from strict laminar flow in the horizontal tube are a result of buoyancy effects." The authors recognize in this manuscript what other researchers have been saying for years – that buoyancy plays a dominant role in flow reactors with residence times of

minutes to hours. They call it a perturbation, but there is nothing small about it, so I suggest that they call it "variations from strict laminar flow".

This point about buoyancy is important, so the authors should include a sentence about it in the abstract and not just in the conclusions.

Conclusions. I disagree with the last statement: "If it had been feasible to mount the flow tube vertically, these effects could have been largely eliminated." Mounting the flow tube vertically will not eliminate these buoyancy effects because it is impossible to eliminate "hot spots" that will drive the buoyancy. Instead, mounting the flow tube vertically will likely enhance the buoyancy effects. By luck or design, this enhancement can occur in a way that effectively isolates the center flow from the walls, although it greatly shortens the residence time. The flow reactor used for the Kang et al. papers was a vertical flow tube. The authors may want to contact the developers of the original PAM chamber to learn what they know about the flow characteristics in vertical flow reactors if they really want to retain this sentence in the manuscript.

Technical corrections / comments:

4. page 9, line 10. Please change "may" to "will".

Figure 11. Are the two dashed lines switched? I would think that the gravitational settling would cause a lower penetration efficiency for larger particles.

---

## Referee Comment (RC2) · Anonymous Referee #2 · 14 Oct 2016

Huang et al. Present a detailed computational fluid dynamics (CFD) model of their laminar flow tube. They highlight the importance of flow-shaping and mixing at the inlet of the reactor and how non-isothermal conditions can substantially impact on e.g. the distribution of residence times (RTD). Computed RTDs were compared with measured residence times of traces gases, especially $O_3$ following pulsed admission to the flow tube. Introduction of eddy-like diffusivity was required to align calculation and observation.

Without presenting anything particularly new or unexpected, Huang et al. compile some useful information concerning the design and construction of high pressure, laminar flow-tubes with long residence times and draw comparison with some previous designs. This paper is a good starting point for anyone looking to construct such an experiment for investigation of atmospheric processes and is thus appropriate for AMT. The manuscript is clearly written and organized, the figures are appropriate and of good quality. The authors may consider the following comments / questions.

There is almost no indication in this manuscript (P1L15, P2L4) about what atmospheric, chemical / physical systems the flow-flow is intended to be used for. Likewise, at the end of the manuscript, apart from mention that some gas-phase organics may be lost at higher rates to the walls, there is little indication of what type of chemical systems may be problematic. This manuscript would benefit from such considerations even if only qualitative.

P2L4   There are many examples of problems associated with increasing radical concentrations to reduce the time scales on which atmospheric processes take place (>days) to make them observable on typical (hours) laboratory time scales. Not all processes are linear in time x concentration. Obvious examples involve reaction terms (e.g. reactions between peroxy radicals) that are quadratic in concentration. This deserves mention.

P3L4   The max. Reynolds number in the conical section is listed here. It would also be useful at this point to list the Reynolds number in the long cylindrical section of the flow tube.

P3L9   The lamps emit mainly at 254 nm. Please indicate the relative intensity (penetrating the jackets and water coolant) of the other lines are. Perhaps a Figure of the lamp emission spectra would be useful.

P5L20 Ezell were not the first to use the shower-head design ?. I'm aware of others (constructed for a similar purpose) that precede this by several years (e.g. Bonn et al., J. Phys. Chem. 106, 2869, 2002).

P11L21 The diffusivity for ozone can be accurately calculated or simply taken from previous experimental determinations found in the literature. Why assume a value of 1e-5 m^2/s?

P17L29 the authors write: The current study indicates that secondary flows can exist in laminar flow tube reactors and affect the fluid dynamics and RTD. This is true but this is not the first time that this fact has been established. Previous users of high-pressure, laminar flow tubes have recognized this fact and done detailed characterization of the effects of mixing and non-laminar conditions by conducting kinetic experiments with well-known reaction systems (see e.g. Donanhue et al. J. Phys. Chem. 100, 5821, 1996). While the pulsed addition of a trace gas will provide an estimate of the RTD, the study of a reaction would provide even more insight as the effects of mixing of trace gases of different diffusivity an wall losses can be assessed.

---

## Referee Comment (RC3) · Anonymous Referee #3 · 21 Oct 2016

The Caltech Photooxidation Flow Tube Reactor – I: Design and Fluid Dynamics Y. Huang et al., Atmos. Meas. Tech. Discuss., doi:10.5194/amt-2016-282, 2016

The authors present a detailed description of the design and characterisation of the new Caltech photooxidation flow tube (CPOT) reactor, using a combination of fluid dynamics calculations to determine the ideal flow behavior within the reactor and experimental characterisations to assess deviations from ideal conditions. The manuscript is well written and the design and results are presented in a clear and logical manner. The manuscript is suitable for publication in AMT. I have only minor comments listed below:

In general the introduction is quite short, and would benefit from a discussion of the various types of flow tubes currently in operation. The authors comment that flow tube

[Figure]

reactors are an alternative to the batch chamber, but do not comment on the different timescales of processes that are often studied with flow tubes or batch reactors – they are often quite different.

The introduction would also benefit from a discussion of the limitations of previous flow tube designs, and any evidence that the fluid dynamics in previous flow tube designs are not well represented by simple models.

Page 2, line 33: Please indicate the section number where the temperature control of the inlet diffuser is described. Page 3, line 9: 'irradiate' to 'irradiation'. Page 3, line 17: Please quantify 'substantial amount of energy'. Page 3, line 30: 'condition' to 'conditions'. Page 4, lines 9-11: Mixture of numbers given as 'two' and '2'. Please be consistent. Page 5, line 8: Please move the references to after 'PAM reactor'. Page 5, line 19 (and elsewhere): 'reactor' should be included wherever 'PAM' is mentioned. Page 6, line 4: Although it is defined in the abstract, please also define 'RTD' here. Page 10, equation 8: Please provide some explanation for the ellipsis in the equation. Page 10, line 11: Please provide some typical values to give a reference point for small values being less than 0.02. Page 15, line 18: Please change 'minute' to 'small'. Page 16, line 5: 'An indealized...'. Page 26, Figure 3: The photograph is not particularly clear. Page 32, Figure 9 caption (end): 'non-diffusion' to 'non-diffusing'.

---

## Referee Comment (RC4) · Anonymous Referee #4 · 27 Oct 2016

The manuscript "The Caltech Photooxidation Flow Tube Reactor – I: Design and Fluid Dynamics" by Huang et al. presents a very detailed description of the design and characterization of a new laminar flowtube reactor (CPOT) for oxidation reactions of vapor molecules and particles at high oxidant levels. The paper focuses on the simulation of fluid dynamics of several fundamental components, such as inlet and exit section, and residence time distributions (RTD) under conditions deviating from the ideal situation. Modeled RTDs are finally compared to experimental RTDs in order to evaluate the performance of the CPOT. Exploring chemical aging of gas phase molecules and particles relevant in the environment and optimizing measurement techniques is certainly of fundamental interest. Given the complexity these kinetics, laboratory experiments on confined and characterized systems in a well-defined environment are required for developing a better understanding of the complex processes occurring in the atmosphere. The manuscript is well written and clearly structured, and although the novelty of the presented material is questionable, it provides a detailed summary of important factors that need to be considered when building a new flow reactor system. One weakness of the presented flow reactor is the high level of oxidant that is required. The authors write in the introduction that high OH exposures are equivalent to multiple days of atmospheric OH concentrations. However, whether high oxidant levels can be extrapolated to atmospherically relevant conditions has yet to be determined. In fact, it is very likely that the chemical reaction pathway might change as a function of reaction time and oxidant concentration. The authors attribute in section 5.1.2 RTD discrepancy to the evaporation of particle-borne water. It would be interesting to see how relative humidity affects CPOT performance.

---

## Referee Comment (RC5) · A. Lambe (Referee) · 15 Nov 2016

Huang and Coggon et al. present theoretical and experimental evaluation of a newly-designed Caltech Photooxidation Flow Tube Reactor (CPOT). The CPOT incorporates components of, and is evaluated against, other recently developed oxidation flow reactor techniques. The authors use COMSOL to conduct CFD simulations of various inlet configurations (e.g. conical diffusers, static mixers) and lamp-induced temperature gradients, and their effects on flow fields. The penetration efficiency ($\eta$) and residence time distributions (RTDs) for vapors ($O_3$, $SO_2$, $H_2O_2$) and particles (polydisperse ammonium sulfate) are measured experimentally. For the vapors that were studied, $\eta = 100\%$; for particles, $\eta \leq 80\%$ at $D_p = 100$ nm. The authors use a Taylor dispersion model to simulate the observed RTDs, and specifically to reproduce behavior of temperature-gradient-induced secondary flows.

Overall, in my opinion this manuscript is well written. The CPOT technique has potential applications for laboratory SOA studies that will presumably (based on title) be examined in related publications to follow. Another, perhaps even more important contribution is the theoretical and modeling framework that is presented which is applicable to other oxidation flow reactor techniques. I would support publication in Atmospheric Measurement Techniques after consideration of my comments below.

Comments

1. Aside from mentioning that the CPOT is equipped with UV-A, UV-B, and mercury lamps, there is no discussion of the photochemical oxidation capabilities of the CPOT despite the statement in Section 2.2 that "quantifying light fluxes for each type of lamp is the prerequisite for performing photochemical experiments in the CPOT." In my opinion, it is critical to provide basic information in this regard in this paper. I recommend supplementing Section 2.2 and Figure 1D by adding a section (or table) summarizing the basic photochemical oxidation capabilities of the CPOT. For example:

| Lamp Type | Primary / Mean Emission $\lambda$ | Minimum actinic flux | Maximum actinic flux | OH precursor(s) | Minimum [OH] | Maximum [OH] |
|---|---|---|---|---|---|---|
| UVC (Hg) | 254 | XX | XX | AA | xx | xx |
| UVB | 305 | YY | YY | BB | yy | yy |
| UVA | 350 | ZZ | ZZ | CC | zz | zz |

2. I think it would also be useful to briefly mention how the lights are used. For example, are only UVA, UVB, or UVC lamps used depending on the experimental goals? Or are combinations of different lamp types used at the same time? How is the UV intensity adjusted and measured?

3. I want to point out that the "straight tube inlet" design portrayed in Figure 3A doesn't incorporate specific characteristics of the PAM reactor, which uses a drilled-out inlet nut on the inside of the front plate combined with a fine mesh screen or a Teflon disc (depending on version; see Figure 1 below) to promote radial mixing. If the goal of this analysis is to represent the specific characteristics of the PAM reactor -- which, for self-serving reasons, I'd be curious to see -- I suggest incorporating additional inlet components shown in the photo to evaluate the effect it has (if any). I can provide necessary specifications. Otherwise, if it is only meant to illustrate a

simplified reactor geometry, it shouldn't be specifically associated with the PAM reactor as is currently done in the Figure 3 caption.

[Figure]

**Figure 1**. PAM reactor "inlet mixer". A mesh screen or Teflon disc is press fit into a nut with drilled-out holes to promote axial mixing.

4.  There are several sections in the paper where extensive sets of equations / derivations are used:
    - Section 4.2, Equations 3-7
    - Section 4.3.1, Equations 10-15
    - Section 4.3.2, Equations 16-22

    While informative and useful for advanced readers, in my opinion, this level of detail is potentially overwhelming for basic readers. Would it be possible to move some of this material to a new Appendix C?

5.  **P14, L9-L11**: The authors state: "The extent of wall deposition of organic vapors in the flow tube reactor requires comprehensive study and will be addressed in a future publication." Isn't this also applicable to the conical diffuser / static mixer described earlier in the paper (P6, L27), which has higher surface-to-volume ratio than "Reactor 1" and "Reactor 2"? I suggest briefly stating somewhere in the manuscript that a limitation to using conical diffusers/ static mixers is losses of sticky organic vapors that may be important SOA precursors.

6.  **P14, L28**: "Note that since we measured the particle size distribution after the static mixer, this loss does not arise from the static mixer." Related to #4, doesn't this imply that there is significant loss of particles through the static mixer? I suggest briefly stating somewhere in the manuscript that a limitation to using static mixers is particle losses. This may unimportant for CPOT-related applications, but is critical in applications of other oxidation flow reactors, particularly when a goal is to measure SOA-to-POA-enhancement ratios (e.g. Ortega et al., 2013; Tkacik et al., 2014; Ortega et al., 2016; Palm et al., 2016, Karjalainen et al., 2016; Simonen et al., 2016).

7.  **Section 4.3**: Please also cite the use of Taylor dispersion modeling to characterize vapor and particle residence time distributions in the PAM and TPOT reactors by Lambe et al. (2011). On a related note, an RTD comparison figure presented by Campuzano-Jost et al. (2016) is reproduced below (Figure 2). Can the authors provide any insight or hypotheses as to how the (seemingly) minor improvement in RTD obtained with the CPOT would influence the corresponding measured properties of secondary organic aerosols produced in both systems?

[Figure]

**Figure 2**. Comparison of theoretical and measured oxidation flow reactor residence time distributions (RTDs) in CPOT and PAM by Campunazo-Jost et al. (2016)

8. **Sect. 4.2.3**: Please define size limits for "diffusive" and "non-diffusive" particles.

9. **Figures 7-8**: I am not certain if these figures are necessary. If they are, I wonder if they could be moved to an appendix/supplement.

10. **Figure 9**: For "100 nm, 99%", "200 nm, 95%", "500 nm, 88%", and "1000 nm, 65%), I assume that the 99%, 95%, 88% and 65% values are penetration efficiencies. This could be made clearer, for example, "100 nm, $\eta$ = 99%". In the open area (white spaces), the authors may also consider adding an "$\eta$ = 0%" label, or modifying the caption to read: "The open space between the dashed curve and the tube wall indicates the region in which particles have deposited on the tube wall **($\eta$ = 0% )**".

11. **Figure 11**: Please indicate in the caption that the penetration efficiency as defined here does not include particle losses in the diffuser/static mixer, and thus represents a lower limit to the pentration efficiency of the entire CPOT.

12. **Figure 13**: The authors might consider mentioning in the legend or the caption that Eq. 14 represents a Taylor dispersion model.

References

A.M. Ortega, D.A. Day, M.J. Cubison, W.H. Brune, D. Bon, J. de Gouw, and J.L. Jimenez. Secondary organic aerosol formation and primary organic aerosol oxidation from biomass-burning smoke in a flow reactor during FLAME-3. *Atmos. Chem. Phys.*, 13, 11551-11571, doi:10.5194/acp-13-11551-2013, 2013.

D. S. Tkacik, A. T. Lambe, S. Jathar, X. Li, A. A. Presto, Y. Zhao, D. R. Blake, S. Meinardi, J. T. Jayne, P. L. Croteau, and A. L. Robinson, Secondary organic aerosol formation from in-use motor vehicle emissions using a Potential Aerosol Mass reactor. *Environ. Sci. Technol.*, 48, 11235-11242, 2014.

B.B. Palm, P. Campuzano-Jost, A.M. Ortega, D.A. Day, L. Kaser, W. Jud, T. Karl, A. Hansel, J.F. Hunter, E.S. Cross, J.H. Kroll, A. Turnipseed, Z. Peng, W.H. Brune, and J.L. Jimenez. In situ secondary organic aerosol formation from ambient pine forest air using an oxidation flow reactor. *Atmospheric Chemistry and Physics*, Atmos. Chem. Phys., 16, 2943-2970, doi:10.5194/acp-16-2943-2016, 2016

A.M. Ortega, P.L. Hayes, Z. Peng, B.B. Palm, W. Hu, D.A. Day, R. Li, M.J. Cubison, W.H. Brune, M. Graus, C. Warneke, J.B. Gilman, W.C. Kuster, J. de Gouw, and J.L. Jimenez. Real-time Measurements of Secondary Organic Aerosol Formation and Aging from Ambient Air in an Oxidation Flow Reactor in the Los Angeles Area. *Atmospheric Chemistry and Physics*, 16, 7411-7433, doi:10.5194/acpd-16-7411-2016, 2016.

P. Karjalainen, H. Timonen, E. Saukko, H. Kuuluvainen, S. Saarikoski, P. Aakko-Saksa, T. Murtonen, M. Dal Maso, E. Ahlberg, B. Svenningsson, W. H. Brune, R. Hillamo, J. Keskinen, and T. Rönkkö. Time-resolved characterization of primary and secondary particle emissions of a modern gasoline passenger car. Atmos. Chem. Phys. Discuss., 16, 8559–8570, doi: 10.5194/acp-16-8859-2016, 2016.

P. Simonen, E. Saukko, P. Karjalainen, H. Timonen, M. Bloss, P. Aakko-Saksa, T. Rönkkö, J. Keskinen, and M. Dal Maso: A New Oxidation Flow Reactor for Measuring Secondary Aerosol Formation of Rapidly Changing Emission Sources, Atmos. Meas. Tech. Discuss., doi:10.5194/amt-2016-300, in review, 2016.

P. Campunazo-Jost, Z. Peng, B. B. Palm, W. Hu, and J. L. Jimenez. Recent OFR work from the Jimenez Group at CU-Boulder, 4th PAM Users Meeting, Portland, OR, USA, 20-Oct-2016.

---

## Author Comment (AC1) · 9 Dec 2016

**Response to reviewers:**

**The Caltech Photooxidation Flow Tube Reactor - I: Design and Fluid Dynamics**

We sincerely thank the four anonymous reviewers and Dr. Andrew Lambe for valuable suggestions. We accept most of the reviewers' suggestions and have made substantial revisions. Here we list the major changes to the current version:

1. We have rewritten the **Introduction** to clarify the goal of building the CPOT and to carry out comparison with existing flow tube reactors.

- 2. We have rewritten and added two new sections in Section 2 Design and Experimental Setup:
  - Section 2.2 Photolytic Environment regarding the quantification of light intensity, details are discussed in Appendix A, and the light spectrum is shown in Fig. 2;
  - Section 2.6 Photochemical Model regarding the construction of a basic photochemical model.

3. We have moved the theoretical contents about diffusive penetration efficiency, Taylor dispersion, and particle motion trajectory that contain a large number of equations to the appendices as **Appendices D**, **E**, and **F**.

4. Section 5.1 Experimental Evaluation of Penetration Efficiency and RTD has been divided into two new sections:

**- Section 5.1 Experimental Evaluation of Penetration Efficiency:**

- a. We have carried out additional experiments to address the loss fraction of gas-phase species to the static mixer as a function of relative humidity in Section 5.1.1, and Fig. 9 is updated with four subplots.
- **b.** We have added a new paragraph and **Fig. 10** in **Section 5.1.2** discussing about the impact of coagulation on particle size distribution.
- Section 5.2 Experimental Evaluation of RTD

5. Section 5.2.1 and 5.2.2 has been emerged as a whole section. In the new version, it is **Section 5.3 Non-ideal Flow in the Reactor**.

6. We have added **Section 6 Photochemical Model** talking about the 1D axially-dispersed plug flow reactor (AD-PFR) photochemical model and **Table 1** summarizing the simulated OH exposure that can be achieved by the reactor.

7. We have rewritten the **Conclusions** focusing on the new findings of this work.

8. We have changed the manuscript title.

With these major modifications, particularly the addition of the photolytic environment and a photochemical model, we believe that this manuscript is significantly stronger and more independent. We now consider this manuscript as a stand-alone publication, as opposed to the original plan of publishing a following publication to further characterize the CPOT.

We decided to remove "I" in the title and retitled the paper as "**The Caltech Photooxidation Flow Tube Reactor - Design, Fluid Dynamics and Characterization**".

Below are our point by point responses (in blue). All the changes are marked in the revised paper. The page and line numbers in the response correspond to the revised version.

**Response to Anonymous Referee #1**

**General comments:**

Huang et al. presents a nice manuscript about the new CalTech Photooxidation Flow Tube Reactor (CPOT), which will enable them to join the dozens of research groups who have been using Oxidation Flow Reactors to study the secondary particles and their properties for oxidation times scales appropriate for the atmosphere for a decade. The manuscript is well written and well organized. It contains lots of nice color figures and drawings and solid analyses. However, there are some errors and omissions which must first be addressed, but when they are, the manuscript will be suitable for AMT.

We thank the reviewer for the valuable information about the PAM reactor and the constructive suggestions on this manuscript.

**Specific comments:**

When designing a flow reactor or chamber of any sort, the designer needs to consider how the flow reactor or chamber will be used. There are always compromises because there is no perfect flow reactor or chamber, particularly for atmospheric pressure. For instance, if you want to relate time in the reaction to reaction time, then you will have enhanced wall interactions because that is how this characteristic is achieved. For a flow tube that is at atmospheric pressure, you must have something to mix the different gases and particles to be studied completely into the carrier air flow. This mixing may be by shower heads or static mixers with lots of surface area, but then you will also have lots of possible surface interactions and losses for gases and particles before they even enter the main flow reactor.

If you want to study ambient aerosol particle chemistry or particle mass yields, then you will want to minimize wall effects and losses, which means minimizing wall interactions, but then the ability to relate time in the flow tube to reaction time may become difficult. If you want only to do laboratory studies, you still need to decide whether you want to look at chemical evolution (so that time in the reactor equals reaction time) or chemical properties that are distorted by wall interactions, such as SOA yields. Then that decision determines if you embrace wall interactions to give you laminar flow or you try to avoid walls and lose the ability to relate time in the chamber to reaction time. Which way the compromises are made depends entirely on the purpose. Of course you can decide to try to do both, and possibly compromise both.

The authors of this manuscript should have an introductory paragraph stating what they intend to study with this CPOT so that the reader can judge whether they have designed it appropriately for that purpose. They should also lay out the strengths and weaknesses of their approach for each type of study they intend to do (e.g., SOA mass yields, SOA chemical properties, time-dependent kinetic studies).

**We have substantially rewritten the Introduction. Please refer to the revised paper.**

2.5 Computational Fluid Dynamics (CFD) Simulations page 4, line 20. Assuming isothermal flow is unrealistic, which the authors say later in the manuscript. The authors should say up front that they are going to try isothermal to see what they get and then will need to modify this assumption later in the manuscript.

**We have added a sentence as "The isothermal assumption will be relaxed subsequently." (P6L7).**

3.1 Injection method. Page 5, line 8. The authors are wrong about the method of injecting air into the PAM chamber. The flow does enter through a tube in the middle, but is quickly diverted by a covered cap with holes drilled sideways to divert the flow toward the outer edges of the chamber. This air is then passed through a fine-mesh screen to break the larger eddies into smaller eddies inside the actual chamber. I suggest that the authors get in touch with the builders of the PAM chamber, so that they know and are thus able to give a correct description of the actual flow in the PAM chamber.

We thank the referee for pointing this out. Together with the photo provided by Dr. Lambe, we now know exactly how the PAM injection system works. To avoid any confusion, we have removed all references to the PAM reactor at this point but kept the discussion of the straight injection style, which is referred as the sudden expansion case in the CFD simulation.

Page 5, line 16. This is not the dead volume to which Lambe et al., are referring. I recommend that the authors contact the designers of the PAM chamber to make sure they understand exactly what the PAM chamber design is, as well as the compromises that were made and why.

We have removed this part and added a discussion in a later section about loss in the inlet part. Please refer to our response to the following comment.

Page 5, line 18. The PAM chamber used in Ortega et al., is basically the same as the one used in Lambe et al.. The only difference is that the entrance plate with the flow diverter is removed so as to completely eliminate the loss of ambient particles and gases on the entrance plate and diverter by having the air flow into the chamber over almost its entire cross section. It is not to "reduce recirculation" in the reactor, as stated in this manuscript, although it might have improved the flow somewhat. The main result was that there was no longer any ambient particle loss in the PAM chamber.

We thank the referee for pointing out this fact. We have removed the sentence in question and discuss briefly the effect of the inlet design on the transmission of reactants in Section 3.1 (**P7L28**): "In addition to the flow field inside the reactor introduced by the inlet design, the transmission of different reactants (i.e. gas-phase species and particles) in the inlet system should also be considered (Karjalainen et al., 2016; Ortega et al., 2013, 2016; Palm et al., 2016; Simonen et al., 2016; Tkacik et al., 2014). Generally, a larger surface area means more interaction between the reactants and the walls, especially for "sticky" molecules. The effect of static mixer on the transmission of gas-phase species will be investigated in Section 5.1."

Section 3.2. Page 6, Can the light from the reactor section get into the diffuser section? Is there ozone in the diffuser section? If so, then shouldn't the authors consider the diffuser section part of the reactor?

The diffuser section is covered by aluminum foil, but still a small amount of light leaks from the open side; since the residence time within the diffuser section is much shorter (30 s), as compared with the average residence time in the flow tube (1520 s), so the diffuser section as part of the reactor can be neglected without any discrepancy.

How does the lack of laminar flow in initial part of the diffuser section affect the particle formation/chemistry that the author may want to study since the diffuser is part of the reactor?

As noted above, given the weak light leakage and the short residence time, the effect of reactions occurring within the diffuser section are expected to be minor.

Section 3.3. The authors did look at issues of convection due to the temperature gradients in CPOT, but not the convection due to "hot spots" along the walls, which are likely to be important, irregular, and possibly transient.

Did the authors measure the temperature distribution of the walls in the CPOT to within 0.1 C? I would assume that there will be hot spots with differences of as much as 0.2 C or more, even with the temperature-control jackets and especially when the lights are on. My guess is that assuming an axial temperature gradient in the non-isothermal case is the best case scenario and that reality will be quite a bit worse.

We could do the following analysis with the Richardson number, but we can illustrate the problem with an even simpler calculation. Simple buoyancy calculations would suggest that air warmed by these 0.2 C hot spots would stir the reactor vertically from bottom to top in less than 10 seconds, which is a much shorter time than the 20-minute residence time. And, because the reactor is a round tube, this vertical motion will get translated into horizontal motion as the rising air hits the top half of the tube. The stirring due to these hots spots is likely to be as important as any other flow

considerations and is probably a major driving force in their eddy diffusion correction. Such eddy diffusion will do a great job of bringing the air in the chamber into close contact with the walls several times, thus increasing wall loss and exchange.

The temperature gradients that the authors determine are important are about the same as the 0.2 C difference used in this simple calculation. Thus, they have every right to be worried about thermal gradients and should include hot spots in their thinking about how convection is going to be a huge problem.

We do not measure the temperature distribution of the wall. Instead, we measure the water temperature into and out of the water jacket, and find a temperature difference of ~0.2 K. We agree with the reviewer's point that the presence of a hot spot is likely to induce secondary flow. We have added the following statement (**P16L28**):

"The spiral flow is more easily established if there are hot spots inside, which can be likely, as the sample ports on the reaction sections are not heat-insulated by the water jacket."

5. Results and Discussion 5.1 Measuring the initial value of the gases and particles downstream of the static mixer does not give a complete picture of particle and gas loss if those particles and gases will be added upstream of the static mixer during the experiments. How much of the gases and particles is lost on the static mixer? My guess is that a significant amount of the SO2 and the ammonium sulfate particles will be lost, as will some of the H2O2. Now, if the authors intend to measure the amounts of gases and particles downstream of the static mixer and use these values as their initial values, then this test would be valid. Is that their intent?

Also, what about any gases / particles that are modified by interactions with the static mixer? They could be important for the chemistry that occurs in the CPOT reaction section.

Even for the way this penetration study was done, my guess is that the authors had to wait a while for the surfaces to acclimate to  $SO_2$  before they were able to get it all through. Did the authors have to wait?

Yes, losses can occur in the static mixer and subsequently in the flow tube itself. We have measured the penetration efficiency of gas molecules and polydispers particles in the static mixer and the flow tube individually. The results are now discussed in **Section 5.1** and **Fig. 9**.

For each penetration study, the data in **Fig. A.1** (**Appendix A**) show that at least 1 h is needed for the system to achieve steady state.

Page14, line 8. This test does not provide evidence that there was negligible interaction with the walls for the gases. All the authors can really say is that there was negligible loss, which could mean that the surfaces were (temporarily?) passivated. How that would change with the lamps on and active chemistry is anyone's guess.

It would be good to know the penetration efficiency for both initial cases – before the static mixer and just after the static mixer. The authors should measure it and report the results in the next version of the manuscript.

Also, were these penetration efficiency tests done under dry conditions? If so, then these are "best case" values, especially for SO2 and  $H_2O_2$ . Unless the authors intend to do all their experiments under dry conditions (which will greatly limit the maximum OH exposure in CPOT and the applicability of the results to the real atmosphere), they should do the penetration efficiency tests under the same conditions as they will use in the experiments. My guess is that the penetration efficiency for SO2 and  $H_2O_2$  will fall rather precipitously.

In the additional experiment, we have investigated the RH-dependent penetration efficiency for both gas-phase species and polydisperse ammonium sulfate (RH smaller than its deliquescence RH). Details are now given in **Section 5.1** and **Fig. 9**.

5.1.2. page 14, line 27. It is doubtful that the 20% particle loss is due to electrostatic loss to the quartz walls. Instead, it is likely that the loss is due to more active convection taking the particles closer to the wall more frequently in the CPOT than the authors think.

Upon further consideration, we agree with the referee. We have removed the statement about the electrostatic loss and state that "The measured maximum penetration efficiency is ~ 80%, indicating a loss of particles, which is likely caused by secondary flow that actively conveys particles closer to the wall. This secondary flow will be discussed in Section 5.3." (P14L21) We also added a subsequent paragraph discussing the impact from coagulation (P14L25).

Page 16, line 19. Why are the eddy diffusivities for ozone and particles different? I would think that they should be the same for small particles.

The two values are, in fact, relatively close. The difference may come from different experimental conditions. Since the eddy-like diffusivity,  $\sim 10^{-4}$  m2 s-1, is much higher than the inherent diffusivity of either gas-phase species and particles, all reactants can be viewed as having the same diffusivity.

Somewhere at the end of the discussion and results, the authors need to compare the characteristics of the CPOT to those of other flow tube reactors, such as the PAM chamber, TPOT, and the Irvine chamber. From the data provided here, CPOT appears to be not much better than any of these other existing chambers in terms of particle transmission or residence time distribution. If you look at the difference in the actual arrival time to that predicted by laminar flow and the full-width and also at the half maximum of the actual peak to that of the laminar flow case, as in Figure 10, and compare those to what is achieved in TPOT and PAM in Lambe et al. Figure 3, you will see that the actual

arrival time is about half that for the laminar case and the FWHM peak width is about double that of the laminar case for all three flow tubes.

Therefore, it is unclear to me if the authors have achieved their goal of having a flow tube that can be used for kinetic studies (in the sense of time in the chamber is equal to reaction time) any better than the existing flow reactors can.

We admit that the CPOT does not perform better in terms of RTD than other currently existing flow reactors. Consequently, we have added the following paragraph at the end of Section 5.2 (**P15L29**): "Overall, the experimental RTD results of both gas-phase species and particles in the CPOT are essentially comparable to those of present flow reactors (Lambe et al., 2011a), given the arrival time and the width of the peak. This discrepancy of the RTDs between the theoretical laminar flow and the real flow indicates the presence of non-ideal flow in the reactor."

However, confirmed by the CFD simulation and the fitting results of the experimental data, we can use the simple parameters from fitting results, which incorporates the effect from fluid field (e.g. the axial dispersion) in the diffusion equation, to simplify the modeling of the photochemical studies (**Section 6**). From this point of view, we do have achieved the goal for kinetic studies.

Conclusions. The authors say in the conclusion that "The measured residence time distribution will enable correction for the unavoidable deviations that do occur." The only way this statement can be made credible is for the authors to add a section that shows how they are going to do this and then demonstrate that it works with real kinetics measurements. Then they need to do an error analysis to show what the uncertainty of their kinetic measurements will be. If they do not wish to do this work for this manuscript, then I suggest that they remove this sentence.

**We have removed this sentence.**

Conclusions. The second-to-last sentence in the manuscript says "Finally, the perturbations from strict laminar flow in the horizontal tube are a result of buoyancy effects." The authors recognize in this manuscript what other researchers have been saying for years – that buoyancy plays a dominant role in flow reactors with residence times of minutes to hours. They call it a perturbation, but there is nothing small about it, so I suggest that they call it "variations from strict laminar flow". This point about buoyancy is important, so the authors should include a sentence about it in the abstract and not just in the conclusions.

We have rewritten substantially the **Abstract** and **Conclusion** sections and now state "As confirmed by the CFD prediction, the presence of a slight deviation from strictly isothermal conditions leads to secondary flows in the reactor that produce deviations from the ideal parabolic laminar flow." in the **Abstract**. Conclusions. I disagree with the last statement: "If it had been feasible to mount the flow tube vertically, these effects could have been largely eliminated." Mounting the flow tube vertically will not eliminate these buoyancy effects because it is impossible to eliminate "hot spots" that will drive the buoyancy. Instead, mounting the flow tube vertically will likely enhance the buoyancy effects. By luck or design, this enhancement can occur in a way that effectively isolates the center flow from the walls, although it greatly shortens the residence time. The flow reactor used for the Kang et al. papers was a vertical flow tube. The authors may want to contact the developers of the original PAM chamber to learn what they know about the flow characteristics in vertical flow reactors if they really want to retain this sentence in the manuscript.

**We agree. We have removed this sentence in the revised version.**

Technical corrections / comments:

4. page 9, line 10. Please change "may" to "will".

**Corrected.**

Figure 11. Are the two dashed lines switched? I would think that the gravitational settling would cause a lower penetration efficiency for larger particles.

Thanks for pointing this out. We have corrected it in the revised version by using words instead of equation numbers.

**Response Anonymous Referee #2**

Huang et al. Present a detailed computational fluid dynamics (CFD) model of their laminar flow tube. They highlight the importance of flow-shaping and mixing at the inlet of the reactor and how non-isothermal conditions can substantially impact on e.g. the distribution of residence times (RTD). Computed RTDs were compared with measured residence times of traces gases, especially O3 following pulsed admission to the flow tube. Introduction of eddy-like diffusivity was required to align calculation and observation.

Without presenting anything particularly new or unexpected, Huang et al. compile some useful information concerning the design and construction of high pressure, laminar flow-tubes with long residence times and draw comparison with some previous designs. This paper is a good starting point for anyone looking to construct such an experiment for investigation of atmospheric processes and is thus appropriate for AMT. The manuscript is clearly written and organized, the figures are appropriate and of good quality. The authors may consider the following comments / questions.

**We thank the reviewer for the valuable comments.**

There is almost no indication in this manuscript (P1L15, P2L4) about what atmospheric, chemical / physical systems the flow-flow is intended to be used for. Likewise, at the end of the manuscript, apart from mention that some gas-phase organics may be lost at higher rates to the walls, there is little indication of what type of chemical systems may be problematic. This manuscript would benefit from such considerations even if only qualitative.

**We have rewritten the Introduction to address these issues. Please refer to the revised paper.**

P2L4 There are many examples of problems associated with increasing radical concentrations to reduce the time scales on which atmospheric processes take place (>days) to make them observable on typical (hours) laboratory time scales. Not all processes are linear in time x concentration. Obvious examples involve reaction terms (e.g. reactions between peroxy radicals) that are quadratic in concentration. This deserves mention.

We thank the referee for pointing this out. We agree and have replaced "concentration  $\times$  time" with "integrated OH exposure" in **P2L17**. We also discuss in detail the OH exposure calculation in the new **Section 6**.

P3L4 The max. Reynolds number in the conical section is listed here. It would also be useful at this point to list the Reynolds number in the long cylindrical section of the flow tube.

We have rewritten the following sentence "Under the typical CPOT flow rate (2 L min-1), the Reynolds numbers at the inlet cone, in the cylindrical section, and at the exit cone are 150, 20, and 450, respectively, well below the transition to turbulent flow." in **P4L13**.

P3L9 The lamps emit mainly at 254 nm. Please indicate the relative intensity (penetrating the jackets and water coolant) of the other lines are. Perhaps a Figure of the lamp emission spectra would be useful.

We have rewritten **Section 2.2** and added a new appendix section (**Appendix A**) addressing the quantitation of irradiation from different lamps. **Figure 2** has been added as the reference for the lamp emission spectra.

P5L20 Ezell were not the first to use the shower-head design. I'm aware of others (constructed for a similar purpose) that precede this by several years (e.g. Bonn et al., J. Phys. Chem. 106, 2869, 2002).

We appreciate this comment. We have added the reference and rephrased this sentence as: "Some flow tube designs address inlet issues using flow management devices, e.g. a spoked-hub/showerhead disk inlet (Bonn et al, 2002; Ezell et al, 2010) that distributes the reactants evenly about the reactor cross-section and provides sufficient mixing (Fig. 4B)." in **P7L19**.

P11L21 The diffusivity for ozone can be accurately calculated or simply taken from previous experimental determinations found in the literature. Why assume a value of 1e-5 m^2/s?

We agree. For reference, the diffusivity of  $O_3$  is  $1.44 \times 10^{-5}$  m2 s-1 and SO2 is  $1.12 \times 10^{-5}$  m2 s-1 (Massman, 1998). The value we choose here,  $1 \times 10^{-5}$  m2 s-1, is the typical order of magnitude of the diffusivity of vapor molecules. By using this value, we want to generalize the simulation to the behavior of typical vapor molecules, but, later on, an exact value can be used, if needed.

P17L29 the authors write: The current study indicates that secondary flows can exist in laminar flow tube reactors and affect the fluid dynamics and RTD. This is true but this is not the first time that this fact has been established. Previous users of high-pressure, laminar flow tubes have recognized this fact and done detailed characterization of the effects of mixing and non-laminar conditions by conducting kinetic experiments with well-known reaction systems (see e.g. Donanhue et al. J. Phys. Chem. 100, 5821, 1996). While the pulsed addition of a trace gas will provide an estimate of the RTD, the study of a reaction would provide even more insight as the effects of mixing of trace gases of different diffusivity and wall losses can be assessed.

We appreciate this information. We agree with the statement that the study of the reaction system also provides insight into the flow mixing effect. We constructed a 1D axially-dispersed plug flow reactor (AD-PFR) photochemical model framework (**Section 2.6**) to simulate OH+SO2 with photolysis of  $H_2O_2$  in the absence of NOx. The fitting results from pulsed RTD experiments are used as parameters in this model. The discrepancy between the PFR model and the AD-PFR model at high reaction rate demonstrates the flow mixing effect on the reaction system. Details can be found in **Section 6**.

**Reference:**

Massman, W. J.: A review of the molecular diffusivities of H2O, CO2, CH4, CO, O3, SO2, NH3, N2O, NO, and NO2 in air, O2 and N2 near STP, *Atmos. Environ.*, 32, 1111-1127, doi:10.1016/S1352-2310(97)00391-9, 1998.

**Response to Anonymous Referee #3**

The authors present a detailed description of the design and characterisation of the new Caltech photooxidation flow tube (CPOT) reactor, using a combination of fluid dynamics calculations to determine the ideal flow behavior within the reactor and experimental characterisations to assess deviations from ideal conditions. The manuscript is well written and the design and results are presented in a clear and logical manner. The manuscript is suitable for publication in AMT. I have only minor comments listed below:

In general the introduction is quite short, and would benefit from a discussion of the various types of flow tubes currently in operation. The authors comment that flow tube reactors are an alternative to the batch chamber, but do not comment on the different timescales of processes that are often studied with flow tubes or batch reactors – they are often quite different.

The introduction would also benefit from a discussion of the limitations of previous flow tube designs, and any evidence that the fluid dynamics in previous flow tube designs are not well represented by simple models.

**We thank the reviewer for these specific comments.**

Page 2, line 33: Please indicate the section number where the temperature control of the inlet diffuser is described.

**Corrected.**

Page 3, line 9: 'irradiate' to 'irradiation'.

We have changed 'irradiate mainly at 254 nm' to 'emit narrow bands at 254 nm and 185 nm'.

Page 3, line 17: Please quantify 'substantial amount of energy'.

Given the measured temperature difference (0.3 K), the circulation flow rate (13 L min-1) and the water heat capacity (4200 J kg-1 K-1), assuming the heat absorption efficiency is about 50%, we roughly estimate a value of 550 W. The working power of the lamp is 45 W, thus the energy to generate heat is about 76% of the total energy consumption, which is a reasonable value.

Page 3, line 30: 'condition' to 'conditions'.

**Corrected.**

Page 4, lines 9-11: Mixture of numbers given as 'two' and '2'. Please be consistent.

**Corrected.**

Page 5, line 8: Please move the references to after 'PAM reactor'.

Since the PAM reactor does not use this injection style, we have removed all the statements about the PAM reactor as well as the references in this section. Please refer to the response to Anonymous Reviewer #1 on Page 4 and comment 3 by Andrew Lambe #5 on Page 20 for details.

Page 5, line 19 (and elsewhere): 'reactor' should be included wherever 'PAM' is mentioned.

Please see the response to the previous comment.

Page 6, line 4: Although it is defined in the abstract, please also define 'RTD' here.

**Corrected.**

Page 10, equation 8: Please provide some explanation for the ellipsis in the equation.

(Note: the equation numbers have been changed.)

The analytical solution to Eqs. D2 - D5 is a superposition of infinite eigenvalue terms. The coefficients inside the exponential part of the 4th, 5th, and 6th terms are -107.6, -174.3, and -256.9, respectively, which is so large that as long as  $\xi$  is not small enough (< 0.02), these terms can be omitted. Theoretically, for small values of  $\xi$ , Eq. 3 and 4 can give the same value of  $\eta$  as long as we can find an infinite series of eigenvalues with sufficient precision.

Page 10, line 11: Please provide some typical values to give a reference point for small values being less than 0.02.

In the book *The Mechanics of Aerosols* (Fuchs, 1964), Fuchs has compared Eq. 3 and 4 in Table 16. For  $\xi$  in the range of 0.01 to 0.1, the discrepancy between the two is within 1%. Thus 0.02 is a safe criterion.

Page 15, line 18: Please change 'minute' to 'small'.

**Corrected.**

Page 16, line 5: 'An idealized ... '.

Thanks for pointing this out. We have removed this sentence in order to merge Section 5.2.1 and 5.2.2 as a whole.

Page 26, Figure 3: The photograph is not particularly clear.

We have adjusted the contrast of the photograph to make it clear.

Page 32, Figure 9 caption (end): 'non-diffusion' to 'non-diffusing'.

Corrected. Figure 9 now becomes Fig. F.2 in the new version.

**Reference:**

Fuchs, N. A.: The Mechanics of Aerosols. Pergamon, New York, 1964.

**Response to Anonymous Referee #4**

The manuscript "The Caltech Photooxidation Flow Tube Reactor – I: Design and Fluid Dynamics" by Huang et al. presents a very detailed description of the design and characterization of a new laminar flowtube reactor (CPOT) for oxidation reactions of vapor molecules and particles at high oxidant levels. The paper focuses on the simulation of fluid dynamics of several fundamental components, such as inlet and exit section, and residence time distributions (RTD) under conditions deviating from the ideal situation. Modeled RTDs are finally compared to experimental RTDs in order to evaluate the performance of the CPOT. Exploring chemical aging of gas phase molecules and particles relevant in the environment and optimizing measurement techniques is certainly of fundamental interest. Given the complexity these kinetics, laboratory experiments on confined and characterized systems in a well-defined environment are required for developing a better understanding of the complex processes occurring in the atmosphere. The manuscript is well written and clearly structured, and although the novelty of the presented material is questionable, it provides a detailed summary of important factors that need to be considered when building a new flow reactor system.

**We thank the reviewer for these valuable comments.**

One weakness of the presented flow reactor is the high level of oxidant that is required. The authors write in the introduction that high OH exposures are equivalent to multiple days of atmospheric OH concentrations. However, whether high oxidant levels can be extrapolated to atmospherically relevant conditions has yet to be determined. In fact, it is very likely that the chemical reaction pathway might change as a function of reaction time and oxidant concentration.

We agree with the reviewer's comment. The enhanced oxidation environment in a flow tube reactor may promote reactions, such as  $RO_2+ RO_2$ , which may change the route of formation of the secondary organic aerosol (SOA). However, this potential weakness is common to all the existing flowtube reactors, rather than one that is particularly associated with the CPOT. A recent study have found no significant discrepancy between compositions of SOA generated in the chamber and the flowtube at current detection precision (Lambe et al., 2015). Whether a molecular-level difference exists between the chamber and the flowtube SOA still remains an open question and is of great interest for the atmospheric chemistry community. We believe that a fundamentally wellcharacterized flowtube reactor, such as the one demonstrated in this work, will be beneficial in answering this type of questions.

The authors attribute in section 5.1.2 RTD discrepancy to the evaporation of particle-borne water. It would be interesting to see how relative humidity affects CPOT performance.

We have removed this possible interpretation since we find no relationship between the relative humidity and penetration efficiency. We attribute the discrepancy to the forced convection to the wall by the secondary flow induced by small temperature differences. We also add a new paragraph (P14L25) and Fig. 10 in Section 5.1.2 discussing about the effect of coagulation on the size distribution.

**Reference:**

Lambe, A., Chhabra, P., Onasch, T., Brune, W., Hunter, J., Kroll, J., Cummings, M., Brogan, J., Parmar, Y., and Worsnop, D.: Effect of oxidant concentration, exposure time, and seed particles on secondary organic aerosol chemical composition and yield, *Atmos. Chem. Phys.*, 15, 3063-3075, 2015.

**Response to Dr. Andrew Lambe #5**

Huang and Coggon et al. present theoretical and experimental evaluation of a newly-designed Caltech Photooxidation Flow Tube Reactor (CPOT). The CPOT incorporates components of, and is evaluated against, other recently developed oxidation flow reactor techniques. The authors use COMSOL to conduct CFD simulations of various inlet configurations (e.g. conical diffusers, static mixers) and lamp-induced temperature gradients, and their effects on flow fields. The penetration efficiency ( $\eta$ ) and residence time distributions (RTDs) for vapors (O3, SO2, H2O2) and particles (polydisperse ammonium sulfate) are measured experimentally. For the vapors that were studied,  $\eta$ = 100%; for particles,  $\eta \leq 80\%$  at Dp = 100 nm. The authors use a Taylor dispersion model to simulate the observed RTDs, and specifically to reproduce behavior of temperature-gradientinduced secondary flows.

Overall, in my opinion this manuscript is well written. The CPOT technique has potential applications for laboratory SOA studies that will presumably (based on title) be examined in related publications to follow. Another, perhaps even more important contribution is the theoretical and modeling framework that is presented which is applicable to other oxidation flow reactor techniques. I would support publication in Atmospheric Measurement Techniques after consideration of my comments below.

We thank Dr. Lambe for the constructive suggestions and details about PAM.

**Comments**

1. Aside from mentioning that the CPOT is equipped with UV-A, UV-B, and mercury lamps, there is no discussion of the photochemical oxidation capabilities of the CPOT despite the statement in Section 2.2 that "quantifying light fluxes for each type of lamp is the prerequisite for performing photochemical experiments in the CPOT." In my opinion, it is critical to provide basic information in this regard in this paper. I recommend supplementing Section 2.2 and Figure 1D by adding a section (or table) summarizing the basic photochemical oxidation capabilities of the CPOT. For example:

| Lamp | Primary/Mean       | Minimum      | Maximum      | OH precursor(s) | Minimum | Maximum |
|------|--------------------|--------------|--------------|-----------------|---------|---------|
| type | Emission $\lambda$ | actinic flux | actinic flux |                 | [OH]    | [OH]    |
| UVC  | 254                | XX           | XX           | AA              | XX      | XX      |
| (Hg) |                    |              |              |                 |         |         |
| UVB  | 305                | YY           | YY           | BB              | YY      | YY      |
| UVC  | 350                | ZZ           | ZZ           | CC              | ZZ      | ZZ      |

We have rewritten **Section 2.2** and added **Appendix A** and **Fig. 2** to illustrate the quantification of light intensity and the spectrum of the lamps. **Table B1** in **Appendix B** shows the photolysis rate under full emission of the three types of lamps. A photochemical model is presented in **Section 6** to

|           | 0                    | H exp a | Atmos. Equiv. b |                        |  |
|-----------|----------------------|--------------------------------------|----------------------------|------------------------|--|
| Lamp type | (mole                | $ec \ cm^{-3} \ s)$                  | (h)                        |                        |  |
|           | PFR                  | AD-PFR corr c  | PFR                        | AD-PFR corr |  |
| Hg vapor  | 8.0×10 11 | 7.3×10 11                 | 222                        | 203                    |  |
| UVB       | 5.4×10 10 | $4.9 \times 10^{10}$                 | 15                         | 13.6                   |  |
| UVA       | 6.0×10 9  | 5.4×10 9                  | 1.7                        | 1.5                    |  |

simulate the OH exposure in the absence of  $NO_x$  condition (the OH precursor is  $H_2O_2$ ). We summarize the simulation results in "**Table 1**: Simulated OH exposure under full light emission

aInput of OH exposure (OHexp) simulation: 1 ppm  $H_2O_2$  and 100 ppb SO2 at RH = 5% and T = 295 K ([H2O2] = 1500 ppm).

bAtmospheric equivalent (Atmos. Equiv.)  $OH_{exp}$  values are also converted to their equivalent hours of OH exposure in the ambient atmosphere, assuming a typical ambient OH concentration of  $1 \times 10^6$ molecule cm-3.

cPFR and AD-PFRcorr are calculated by Eqs. (6) and (7), respectively."

2. I think it would also be useful to briefly mention how the lights are used. For example, are only UVA, UVB, or UVC lamps used depending on the experimental goals? Or are combinations of different lamp types used at the same time? How is the UV intensity adjusted and measured?

We have rewritten **Section 2.2**. In the first paragraph we state how the lights are used, including the intensity adjustment and the usage based on the particular experimental goals. In the second paragraph, we briefly explain how we measure the light intensity, with details in **Appendix A**.

3. I want to point out that the "straight tube inlet" design portrayed in Figure 3A doesn't incorporate specific characteristics of the PAM reactor, which uses a drilled-out inlet nut on the inside of the front plate combined with a fine mesh screen or a Teflon disc (depending on version; see Figure 1 below) to promote radial mixing. If the goal of this analysis is to represent the specific characteristics of the PAM reactor -- which, for self-serving reasons, I'd be curious to see -- I suggest incorporating additional inlet components shown in the photo to evaluate the effect it has (if any). I can provide necessary specifications. Otherwise, if it is only meant to illustrate a simplified reactor geometry, it shouldn't be specifically associated with the PAM reactor as is currently done in the Figure 3 caption.

Figure 1. PAM reactor "inlet mixer". A mesh screen or Teflon disc is press fit into a nut with drilled-out holes to promote axial mixing.

We thank Dr. Lambe for providing this figure. To avoid any confusion, we have removed all the statements that describe the specifics of the PAM reactor regarding the "straight tube inlet" injection style. We retain Fig. 3A as the sudden expansion  $(90^\circ)$  case that is discussed in Section 3.2.

4. There are several sections in the paper where extensive sets of equations / derivations are used:

- Section 4.2, Equations 3-7
- Section 4.3.1, Equations 10-15
- Section 4.3.2, Equations 16-22

While informative and useful for advanced readers, in my opinion, this level of detail is potentially overwhelming for basic readers. Would it be possible to move some of this material to a new Appendix C?

**We have moved these developments from the main text into Appendices D, E, and F.**

5. **P14, L9-L11**: The authors state: "The extent of wall deposition of organic vapors in the flow tube reactor requires comprehensive study and will be addressed in a future publication." Isn't this also applicable to the conical diffuser / static mixer described earlier in the paper (P6, L27), which has higher surface-to-volume ratio than "Reactor 1" and "Reactor 2"? I suggest briefly stating somewhere in the manuscript that a limitation to using conical diffusers/ static mixers is losses of sticky organic vapors that may be important SOA precursors.

**We have included a short summary at the end of Section 3.1 concerning the large surface area related loss (**P7L28**):**

"In addition to the flow filed inside the reactor introduced by the inlet design, the transmission of different reactants (i.e. gas-phase species and particles) in the inlet system should also be considered (Karjalainen et al., 2016; Ortega et al., 2013, 2016; Palm et al., 2016; Simonen et al., 2016; Tkacik et al., 2014). Generally, a larger surface area means more interaction between the reactants and the walls, especially for "sticky" molecules. The effect of static mixer on the transmission of gas-phase species will be investigated in Section 5.1."

6. **P14, L28**: "Note that since we measured the particle size distribution after the static mixer, this loss does not arise from the static mixer." Related to #4, doesn't this imply that there is significant loss of particles through the static mixer? I suggest briefly stating somewhere in the manuscript that a limitation to using static mixers is particle losses. This may unimportant for CPOT-related applications, but is critical in applications of other oxidation flow reactors, particularly when a goal

is to measure SOA-to-POA-enhancement ratios (e.g. Ortega et al., 2013; Tkacik et al., 2014; Karjalainen et al., 2016; Ortega et al., 2016; Palm et al., 2016; Simonen et al., 2016).

**Addressing this comment is incorporated in the response to the previous point.**

7. Section 4.3: Please also cite the use of Taylor dispersion modeling to characterize vapor and particle residence time distributions in the PAM and TPOT reactors by Lambe et al. (2011). On a related note, an RTD comparison figure presented by Campuzano-Jost et al. (2016) is reproduced below (Figure 2). Can the authors provide any insight or hypotheses as to how the (seemingly) minor improvement in RTD obtained with the CPOT would influence the corresponding measured properties of secondary organic aerosols produced in both systems?

Figure 2. Comparison of theoretical and measured oxidation flow reactor residence time distributions (RTDs) in CPOT and PAM by Campuzano-Jost et al. (2016).

We have added the following statement in Section 4.3.1 about Taylor dispersion modeling (**P12L17**): "In the PAM reactor (Lambe et al., 2011a), the Taylor dispersion criteria does not strictly meet the working conditions, however, the two flow regime fitting results suggest that two types of flow may exist in the reactor: a direct flow with minor dispersion and a secondary recirculation flow with significant dispersion."

We appreciate Dr. Campuzano-Jost for compiling the RTD data from different types of flow reactor. We also thank Dr. Jimenez for making an update on this figure, proving that the RTD of CPOT looks better. For non-ideal reactors, RTD can provide insights to the fluid field. As confirmed by our CFD simulations and Taylor dispersion fittings, we have successfully simplified the coupling of RTD with the kinetic study, i.e. the convection-diffusion ordinary differential equation (ODE). This makes the interpretation of data more reliable. Please refer to Section 6 for the details.

8. Sect. 4.2.3: Please define size limits for "diffusive" and "non-diffusive" particles.

Since particles always exhibit Brownian diffusivity, to avoid confusion, we note two regimes, i.e. diffusion regime (diameter smaller than ~80 nm) and settling regime (diameter larger than ~80 nm), and kept consistent with this expression in the context (**P13L14**).

9. **Figures 7-8**: I am not certain if these figures are necessary. If they are, I wonder if they could be moved to an appendix/supplement.

We have moved Figs. 8 and 9 to **Appendix F**, but kept Fig. 7 (in the new version it becomes **Fig.** 8) since it relates directly to the two regimes for the discussion about particles.

10. **Figure 9**: For "100 nm, 99%", "200 nm, 95%", "500 nm, 88%", and "1000 nm, 65%), I assume that the 99%, 95%, 88% and 65% values are penetration efficiencies. This could be made clearer, for example, "100 nm,  $\eta = 99\%$ ". In the open area (white spaces), the authors may also consider adding an " $\eta = 0\%$ " label, or modifying the caption to read: "The open space between the dashed curve and the tube wall indicates the region in which particles have deposited on the tube wall ( $\eta = 0\%$ )".

We have modified the title of each panel and the caption of the figure. The new figure now is in **Appendix F** as **Fig. F.2**.

11. **Figure 11**: Please indicate in the caption that the penetration efficiency as defined here does not include particle losses in the diffuser/static mixer, and thus represents a lower limit to the penetration efficiency of the entire CPOT.

We have replaced this figure with a more comprehensive one, showing the penetration efficiency of gas-phase species and particles (**Fig. 9** in the revised paper).

12. **Figure 13**: The authors might consider mentioning in the legend or the caption that Eq. 14 represents a Taylor dispersion model.

Done.

Manuscript prepared for Atmos. Meas. Tech. with version 2016/01/22 8.09 Copernicus papers of the LATEX class copernicus.cls. Date: 8 December 2016

**$\mathcal{D}$**

**The Caltech Photooxidation Flow Tube Reactor - Design, Fluid Dynamics and Characterization**

Y. Huang1,\*, M. M. Coggon2,a,\*, R. Zhao2, H. Lignell2,b, M. U. Bauer2, R. C. Flagan1,2, and J. H. Seinfeld1,2

1Department of Environmental Science and Engineering, California Institute of Technology, Pasadena, CA, USA 2Division of Chemistry and Chemical Engineering, California Institute of Technology, Pasadena, CA, USA aNow at CIRES, University of Colorado, and NOAA Earth System Research Laboratory, Boulder, CO, USA bNow at South Coast Air Quality Management District, Diamond Bar, CA, USA \*These authors contributed equally to this work.

Correspondence to: J. H. Seinfeld (seinfeld@caltech.edu)

**Abstract.** Flow tube reactors are widely employed to study gas-phase atmospheric chemistry and secondary organic aerosol formation. The development of a new laminar-flow tube reactor, the Caltech PhotoOxidation flow Tube (CPOT), intended for the study of gas-phase atmospheric chemistry and secondary organic aerosol (SOA) formation, is reported here. The present work addresses the reactor design based on fluid dynamical characterization and the fundamental behavior of vapor molecules

- 5 and particles in the reactor. The design of the inlet to the reactor, based on computational fluid dynamics (CFD) simulations, comprises a static mixer and a conical diffuser to facilitate development of a characteristic laminar flow profile. To assess the extent to which the actual performance adheres to the theoretical CFD model, residence time distribution (RTD) experiments are reported with vapor molecules ( $O_3$ ) and sub-micrometer ammonium sulfate particles. As confirmed by the CFD prediction, the presence of a slight deviation from strictly isothermal conditions leads to secondary flows in the reactor that produce
- 10 deviations from the ideal parabolic laminar flow. The characterization experiments, in conjunction with theory, provide a basis for interpretation of atmospheric chemistry and secondary organic aerosol studies to follow. A 1D photochemical model within an axially dispersed plug flow reactor (AD-PFR) framework is formulated to evaluate the oxidation level in the reactor. The simulation indicates that the OH concentration is uniform along the reactor, and an OH exposure (OHexp) ranging from  $\sim 10^9$ to  $\sim 10^{12}$  molecules cm-3 s can be achieved from photolysis of H2O2. A method to calculate OHexp with a consideration for
- 15 the axial dispersion in the present photochemical system is developed.

**1** Introduction**

Experimental evaluation of atmospheric chemistry and aerosol formation is typically carried out in laboratory reactors. Such reactors comprise both chambers and flow reactors. The flow tube reactor has emerged as a widely-used platform (Bruns et al., 2015; Chen et al., 2013; Ezell et al., 2010; Kang et al., 2007, 2011; Karjalainen et al., 2016; Keller and Burtscher, 2012;

Khalizov et al., 2006; Lambe et al., 2011a, b, 2012, 2015; Li et al., 2015; Ortega et al., 2013, 2016; Palm et al., 2016; Peng et al., 2015, 2016; Simonen et al., 2016; Tkacik et al., 2014).

The flow tube reactor is generally operated under steady-state conditions. An attribute of the flow tube reactor is that, by control of the inlet concentration and oxidation conditions, it is possible to simulate atmospheric oxidation under conditions

- 5 equivalent to multiple days of atmospheric exposure with a reactor residence time over a range of minutes. It should be noted that the chemistry occurring in such a highly oxidizing environment may differ from that in the atmosphere and batch chamber, even though no discrepancy between the components of the SOA generated in the flow tube reactor and the batch chamber has yet to be reported (Lambe et al., 2015). Moreover, under the steady state operating conditions, it is possible to accumulate sufficient products for detailed analytical evaluation. Key factors relevant to atmospheric processes, such as gas-phase kinetics
- 10 (Donahue et al., 1996; Howard, 1979; Thornton and Abbatt, 2005), nucleation rates (Mikheev et al., 2000), uptake coefficients of vapors on particles (Matthews et al., 2014), and heterogeneous reactions on particle surfaces (George et al., 2007), can be evaluated via flow tube studies.

Since the concept of Potential Aerosol Mass (PAM) was proposed, the PAM reactor, operated as a flow tube reactor, has been widely used in laboratory and field studies of SOA formation (Chen et al., 2013; Kang et al., 2007, 2011; Keller and

- 15 Burtscher, 2012; Kroll et al., 2009; Lambe et al., 2011a, 2012, 2015; Ortega et al., 2016, 2013; Palm et al., 2016; Slowik et al., 2012; Smith et al., 2009). A powerful attribute of the PAM and subsequent flow reactors is the capability to generate hydroxyl radical (OH) levels that lead to integrated OH exposure ranging as high as  $\sim 10^{12}$  molecules cm-3 s, at which it is possible to simulate atmospheric oxidation conditions comparable to those occurring over  $\sim 1$  week. Chemical kinetic modeling studies have investigated the free radical chemistry in the oxidation flow reactor (OFR) (e.g., Li et al., 2015; Peng et al., 2015, 2016).
- 20 Flow tube designs vary in dimension, detailed construction, and strategy for generating the oxidizing environment. Each specific design aspect of a flow reactor can significantly affect both the fluid dynamics and the chemistry within the reactor. For example, the design of the inlet to the reactor determines the extent of initial mixing of the reactants as well as the development of concentration profiles in the reactor. The classical flow tube for gas-phase kinetic measurements employs a movable inlet in the axial position surrounded by a carrier gas to achieve the flexibility in varying reaction time (Howard,
- 25 1979). The wavelength-dependent radiation source determines the choice of oxidants that initiate free radical chemistry. In the atmosphere, the ubiquitous oxidant OH is generated largely by the reaction of  $H_2O$  with  $O(^1D)$ , which is produced by the photolysis of  $O_3$  at wavelengths < 320 nm. In the flow reactor, a variety of OH generation strategies exist. One option is to use blacklights that center around 350 nm to gently photolyze OH precursors such as  $H_2O_2$ , HONO, and CH3ONO. The material of the flow tube determines the placement of radiation sources. For example, the PAM reactor described by Kang et al. (2007)
- 30 is constructed of Teflon which is transparent to UV radiation; consequently, the UV lamps that drive the photochemistry can be positioned outside the reactor itself. By contrast, another class of flow reactors is constructed of aluminum, for which the UV lamps must be positioned inside the reactor itself (Li et al., 2015; Ezell et al., 2010). Characterization of the behavior of the flow tube reactor requires ideally a combination of flow and residence time modeling and experiment, chemical kinetic modeling and experiment, and modeling and experimental measurement of interactions of vapor molecules and particles with
- 35 reactor walls.

We present here the development and characterization of the Caltech Photooxidation flow Tube reactor (CPOT). The CPOT has been constructed as a complement to the Caltech 24 m3 batch chambers (Bates et al., 2014, 2016; Schilling et al., 2015; Hodas et al., 2015; Loza et al., 2013, 2014; McVay et al., 2014, 2016; Nguyen et al., 2014, 2015; Schwantes et al., 2015; Yee et al., 2013; Zhang et al., 2014, 2015) in carrying out studies of SOA formation resulting from the oxidation of volatile

5 organic compounds (VOCs) by oxidants OH, O3, and NO3 over time scales not accessible in a batch chamber. Owing to its steady-state operation, the CPOT also affords the capability to collect sufficient quantities of SOA generated in the reactor for comprehensive composition determination by off-line mass spectrometry.

While the reactor itself is not unlike a number of those already developed and cited above, we endeavor here to describe in some detail the theoretical/experimental characterization of the reactor. Using computational fluid dynamics (CFD) simu-

10 lations, we describe the design and characterization of the CPOT. We highlight fundamental consideration of the design of a laminar flow tube reactor, including methods of injection of gases and particles, the behavior of vapor molecules and particles in the reactor, and effects of non-isothermal conditions on the flow in the reactor. We evaluate the extent to which the fluid dynamics modeling agrees with experimental residence time distribution (RTD) measurements.

Experimental measurements of SOA formation in laboratory Teflon chambers are influenced by deposition of both particles and vapors to the chamber walls, and evaluation of the SOA yield from VOC oxidation must take careful accounting for such wall losses (e.g., Zhang et al., 2014; Nah et al., 2016a, b). We seek to assess the extent to which both vapor and particle depo-

- sition onto the entrance region and quartz wall of the flow tube is influential in flow tube reactor studies. While experimental measurements of these processes will be presented in future studies, the transport modeling presented here provides a basis for evaluating the effect of reactor surfaces on experimental measurements of atmospheric chemistry and SOA formation.
- A photochemical kinetic model is formulated to simulate OH production in the reactor. Typically, at steady state, the flow tube reactor gives only one data point under a specific condition. Such a model is essential in evaluating oxidation data in the reactor since the model predicts how the reactants evolve along the reactor. Generally, the ideal plug flow reactor (PFR) framework is used in the modeling of a flow tube system (Li et al., 2015; Peng et al., 2015, 2016). For a non-ideal flow reactor, the axially-dispersed plug flow reactor (AD-PFR) framework couples the RTD with the chemical reaction system. The axial
- 25 dispersion plays the role of backward and forward mixing of the reactants, smoothing the concentration gradients. By the comparison between AD-PFR and PFR models, we will show how the non-ideal flow reactor impacts the data interpretation and suggest a method for correction.

**2 Design and Experimental Setup**

**2.1 CPOT Reactor**

30 The CPOT comprises three sections: the inlet section, the main reaction section, and the outlet section (Fig. 1A). The inlet consists of two components - the static mixer and the conical diffuser (Fig. 1B). The static mixer is designed to thoroughly mix reactant streams, whereas the diffuser serves to expand the mixed flow to the diameter of the reaction section while maintaining an idealized laminar flow profile. The static mixer is constructed of stainless steel and consists of 12 helical

elements (StaticMixCo, NY). The Pyrex glass diffuser section expands from an inner diameter of 1.6 cm to 15 cm at an angle of 15°. The diffuser angle was chosen based on CFD simulations in order to minimize flow separation and recirculation. Detailed design of the inlet section is discussed in Section 3.

The CPOT reaction section consists of two 1.2 m × 17 cm ID cylindrical quartz tubes surrounded by an external water jacket (1 cm thickness) and flanged together with clamps and chemically resistant o-rings. Four ports along the reactor axis allow sampling of the reactor contents at different residence times. A transition cone at the end of the reactor concentrates the reactants into a common sampling line that can be split among multiple instruments; thus, samples extracted at the end of the reactor of the end of the reactor represent the so-called cup-mixed average of the entire reactor cross section. This design is similar to the exit cone of the UC Irvine flow tube reactor (Ezell et al., 2010). The Pyrex glass exit cone gradually reduces the diameter of the reactor 10 from 15 cm to 0.72 cm at an angle of 15°. Similar to the inlet diffuser, the exit cone is temperature-controlled (Section 2.3).

The CPOT is designed to operate under laminar flow. The essential dimensionless group that differentiates laminar vs. turbulent flow is the Reynolds number,  $\text{Re} = \frac{\rho UD}{\mu}$ , where  $\rho$  is the fluid density, U is a characteristic velocity of the fluid,  $\mu$  is the fluid viscosity, and D is the tube diameter. For cylindrical tubes, the flow is considered laminar when Re < 2100. Under the typical CPOT flow rate (2 L min-1), the Reynolds numbers at the inlet cone, in the cylindrical section, and at the exit cone are 150, 20, and 450, respectively, well below the transition to turbulent flow.

**2.2 Photolytic Environment**

15

The reactor is housed within a  $51 \times 51 \times 300$  cm chamber containing 16 wall-mounted UV lamps. The arrangement of the lamps is outlined in Fig. 1D. Light intensity is adjustable (0, 25%, 50%, 75%, and 100%), and the UV spectrum can be set to a specific wavelength range with the installation of various T12 UV lamps, including Hg vapor lamps (emit narrow bands at

20 254 nm and 185 nm), UVB lamps (polychromatic irradiation centered at 305 nm) and UVA lamps (polychromatic irradiation centered at 350 nm), based on the experimental goals.

Quantification of light fluxes for each type of lamp is the prerequisite for performing photochemical experiments. A challenge associated with quantifying photon fluxes is that the flux emitted by the lamps is not necessarily that perceived by a molecule inside the reactor. Attenuation of photon fluxes can potentially arise from: 1) attenuation by the quartz wall and the

- water jacket surrounding the experimental sections; 2) reflection and/or refraction of light inside the chamber; and 3) absorption of light by gas-phase molecules (e.g. absorption of the 185 nm band by O2 molecules). To overcome this challenge, we employ a method combining direct measurements and gas-phase chemical actinometry, where the directly recorded emission spectra are adjusted to the the observed photolysis rate of NO2 ( $j_{NO_2}$ ). The advantage of this method is that the actual output spectra of the lamps are used, since the quantification of fluxes is based on what the molecules perceive inside the reactor. The
- 30 water coolant in the jacket surrounding the tube is transparent at the UV wavelengths of interest, with the exception that it absorbs at the 185 nm band emitted by the Hg vapor lamps. Although the general UV cutoff of water is at 190 nm, we observed formation of 60 ppb of  $O_3$  with a 2 L min-1 flow rate under the full power of the Hg vapor lamp. The radiation intensity at 185 nm that penetrates into the reaction section is calculated to be about  $10^{-5}$  of that at 254 nm. The photon fluxes in the CPOT

**from the three types of lamps are shown in Fig. 2 with a detailed description of the determination of photon fluxes provided in Appendix A.**

**2.3 **Temperature Control in the Reaction Section**

At full photolytic intensity, the lamps generate as much as 550 W of heat. To maintain a constant temperature and minimize

[revised manuscript text omitted]

**Photochemical Model** 2.6**

**While the CFD simulation serves as a comprehensive method to understand the fluid dynamics, it is not efficient to solve a complex chemical kinetic system within this framework. A simplified 1D axial-dispersion photochemical model based on the RTD measurement is built here to evaluate the oxidation level. The mechanism presented here is that in the absence of $NO_r$ .**

- The oxidation of 100 ppb SO2 by the OH radical is studied.  $H_2O_2$  at 1 ppm serves as the OH precursor. The three types of UV 25 lamps are considered, sequentially, to investigate the effect of the radiation source on OH exposure. Each simulation is carried out at RH = 5% and T = 295 K (corresponding to [H2O] = 1500 ppm). The case in the absence of H2O2 input is also simulated to check the background OH level. Reactions of the full mechanisms and the rate coefficients including photolysis rate under different lamps that are necessary for the chemical kinetic modeling are listed in Appendix B.
- 30 The AD-PFR model setup is used in the present study and described in Appendix C. The Danckwerts boundary condition is employed to ensure the flux continuity at both the inlet and outlet (Davis and Davis, 2003). This model system is solved in MATLAB (R2015b) by a boundary value problem solver byp4c. The PFR model is run simultaneously as a comparison. No wall interaction and new particle formation are considered in the models. The result will be discussed in Section 6.

**3** Design of the Flow Tube Reactor**

Essential elements of the design of a flow tube reactor are: (1) the manner by which reactants are introduced into the reactor; (2) the nature of the flow inside the reactor; (3) the type and location of the radiation source relative to the reactor itself; and (4) the management of heat generation owing to the radiation source. The first two correspond to the inlet section design, while the latter two address the problem of possible non-isothermal conditions in the reaction.

**3.1 Injection Method**

5

A number of possible arrangements exist to introduce material into a flow tube reactor (Fig. 4). The nature of the injection manifold has the potential to profoundly affect the flow profile in the subsequent reaction section. In the case of a laminar flow reactor, it is desirable to minimize such "end effects" in order to establish parabolic flow quickly within the reaction section; otherwise, phenomena such as jetting and recirculation have the potential to impact flow patterns throughout the entire

- 10 section; otherwise, phenomena such as jetting and recirculation have the potential to impact flow patterns throughout the entire reactor. Figure 4A depicts the simplest injection method, by which vapor and particles are introduced into the reaction section through a short injection tube. While a benefit of this design is its simplicity, with this mode of injection, it is challenging to distribute reactant mixtures evenly across the reactor cross section. We tested this inlet method on a cylindrical Pyrex glass tube and visualized the flow pattern by the injection of smoke (Fig. 4A). With flow controlled by a vacuum line attached to
- 15 the exit section, the gas-particle mixture is pulled into the reaction tube at a rate that is dictated by mass conservation. Smoke visualization studies illustrate that the mixture concentrates in a plug at the center of the reactor. This "fire hose" effect arises from the enhanced velocity at the exit of the injection tube ( $U_{avg, injection}$ ). Such flow behavior is typical for that occurring with a sudden expansion (Bird et al., 2007).

Some flow tube designs address inlet issues using flow management devices, e.g. a spoked-hub/showerhead disk inlet (Bonn

- et al., 2002; Ezell et al., 2010) that distributes the reactants evenly about the reactor cross-section and provides sufficient mixing (Fig. 4B). Even when reactants are introduced gently into the tube, an axial distance is still required for the flow to develop to the characteristic parabolic laminar flow profile. This entrance length,  $L_{entr}$ , is estimated to be 0.035*D*Re (Bird et al., 2007). The inlet section should be designed with a sufficient entrance length  $L_{entr}$  to ensure the development of the laminar profile prior to the reaction section.
- In the CPOT, reactants are injected via a conical diffuser (Fig. 4C) which has the advantage of gradually decreasing the velocity, thereby assisting with the formation of the laminar parabolic profile. The employment of a diffuser cone essentially replaces  $L_{entr}$ , and a parabolic profile is fully developed when the reactants reach the reaction section.

In addition to the flow field inside the reactor introduced by the inlet design, the transmission of different reactants (i.e. gasphase species and particles) in the inlet system should also be considered (Karjalainen et al., 2016; Ortega et al., 2013, 2016;

30 Palm et al., 2016; Simonen et al., 2016; Tkacik et al., 2014). Generally, a larger surface area means more interaction between the reactants and the walls, especially for "sticky" molecules. The effect of static mixer on the transmission of gas-phase species will be investigated in Section 5.1.

**3.2 Angle of the Diffuser**

[revised manuscript text omitted]

---

## Short Comment (SC1) · 11 Dec 2016

In one of the reviews of this paper, a figure comparing residence time distributions of several reactors was shown. The figure had originally been presented by our group at the recent PAM Users meeting in Oct. 2016 (https://sites.google.com/site/pamwiki/home/4th-pam-users-meeting). We have since realized that one of the traces contained an error. For the record we reproduce below a proper comparison between the PAM and Caltech flow reactors, as well as the theoretical laminar profile:

[Figure]

**Figure 1**. Comparison of the probability density functions of residence times for different flow reactors, as a function of normalized residence time.

This comparison does not affect the conclusions of the present paper and should not delay its publication.

The plot above is available in Igor format at the following web page, with the hope that it can serve as a point of comparison for RTDs measured by others for these and other reactors:

https://sites.google.com/site/pamwiki/hardware/estimation-equations